# Genetic Mining of Newly Isolated Salmophages for Phage Therapy

**DOI:** 10.3390/ijms23168917

**Published:** 2022-08-10

**Authors:** Julia Gendre, Mireille Ansaldi, David R. Olivenza, Yann Denis, Josep Casadesús, Nicolas Ginet

**Affiliations:** 1Laboratoire de Chimie Bactérienne, (UMR7283)-CNRS/Aix-Marseille Université, 13009 Marseille, France; 2Departamento de Genética, Facultad de Biologia, Universidad de Sevilla, 41012 Sevilla, Spain; 3Institut de Microbiologie de la Méditerranée, (FR3479)-CNRS/Aix-Marseille Université, 13009 Marseille, France

**Keywords:** bacteriophages, *Salmonella*, phage therapy, biocontrol, genomic analysis, functional annotation, PHROG database

## Abstract

*Salmonella enterica*, a Gram-negative zoonotic bacterium, is mainly a food-borne pathogen and the main cause of diarrhea in humans worldwide. The main reservoirs are found in poultry farms, but they are also found in wild birds. The development of antibiotic resistance in *S. enterica* species raises concerns about the future of efficient therapies against this pathogen and revives the interest in bacteriophages as a useful therapy against bacterial infections. Here, we aimed to decipher and functionally annotate 10 new Salmonella phage genomes isolated in Spain in the light of phage therapy. We designed a bioinformatic pipeline using available building blocks to de novo assemble genomes and perform syntaxic annotation. We then used genome-wide analyses for taxonomic annotation enabled by vContact2 and VICTOR. We were also particularly interested in improving functional annotation using remote homologies detection and comparisons with the recently published phage-specific PHROG protein database. Finally, we searched for useful functions for phage therapy, such as systems encoded by the phage to circumvent cellular defenses with a particular focus on anti-CRISPR proteins. We, thus, were able to genetically characterize nine virulent phages and one temperate phage and identify putative functions relevant to the formulation of phage cocktails for *Salmonella* biocontrol.

## 1. Introduction

The massive and often indiscriminate use of antibiotics for decades in human and animal healthcare as well as in agriculture gives rise to antibiotic multi-resistance in bacterial pathogens; this has become a major concern worldwide as it leads to an increasing number of therapeutic failures. According to current projections reported by the World Health Organization (WHO), infectious diseases could once again become one of the leading causes of death in the world by 2050 and cause dramatic damage to the world economy [1]. The “One Health” concept recognizes the interdependency between the environment, human health and animal health; it is promoted by various international organizations such as the WHO, the Organization for Animal Health (OIE), and the Food and Agriculture Organization (FAO) (see for a definition the FAO website at https://www.fao.org/one-health/en) (accessed on 14 June 2022). One Health aims, among other objectives, to promote the rational use of antibiotics and the development of alternative strategies to tackle bacterial infections.

Bacteriophages (or phages) are viruses that infect bacteria and thus represent, as anti-bacterial agents, a promising alternative (or complement) to antibiotics. A given bacteriophage specifically infects a more or less narrow range of bacteria, sometimes down to the species or serotype level. Furthermore, bacteriophages are very diverse and generally easy to isolate and produce. Bacteriophages were independently discovered during the First World War by Frederick W. Twort in 1915 [2] and by Félix d’Hérelle in 1917 [3], who coined the term “bacteriophage” (see Félix d’Hérelle seeding article presented in English by Dr. Roux [4]). Very rapidly in 1918, when antibiotics did not even exist, Félix d’Hérelle foresaw how bacteriophages could be used against bacterial infections and successfully utilized them to treat dysentery [5]. Phage therapy was born and developed to treat both humans and animals. However, in the 1940s, phage therapy was supplanted by the emergence of antibiotics and was abandoned in the Western world, but it was kept in use until today in the former Eastern Bloc countries and former USSR member states [6]. Nowadays, there is a renewed interest worldwide for bacteriophages and phage therapy in the search for an alternative to antibiotics to tackle multi-resistant bacteria and develop targeted antibacterial therapies.

*Salmonella enterica* is a Gram-negative zoonotic bacterium and a major cause of diarrhea worldwide. Avian breeding farms are the main reservoirs for the *Salmonella* species, transmitted to humans by food, especially eggs. In the concept of “One Health”, treating these reservoirs would have a positive impact on human health. Isolation and characterization of phages targeting *Salmonella* are thus important to develop *S. enterica* biocontrol strategies and prevent massive antibiotic resistance in animal reservoirs. Phage therapy relies essentially on the formulation of a cocktail, regrouping several different phages in order to avoid the emergence of bacterial resistance, especially cross-resistance, and to eventually target as wide a range of bacterial serotypes as possible. Another beneficial property of bacteriophage cocktails is that they can be designed to target specific pathogens, leaving the rest of the microbiota unaffected, which is not the case with antibiotics and their indiscriminate action against a wide range of bacteria. A recent and relevant study by Nale et al. highlights the synergistic effects of a three-phage cocktail against prevalent *Salmonella* serotypes on poultry and pigs [7]. Using *Galleria mellonella* as a model that correlates to large-scale animal models, the authors proved that, in combination, their 3 phages could lyse about 99.97% of the 22 serotypes they investigated. They also showed in vivo in the *Galleria mellonella* model that their cocktail was very efficient as a prophylactic agent. Closer to human health issues is the recent report of the successful treatment of a patient infected by a multi-drug resistant *Mycobacterium abscessus* strain by a patient-tailored three-phage cocktail [8]. However, the authors cautiously warn that a generalized approach is far from being a reality.

Just like antibiotherapy, phage therapy collides with bacterial resistance to phage infection. Its success relies on the evaluation of such a probability, implying that we should not reproduce the errors we made with antibiotics. Indeed, the co-evolution of bacteria and their viruses for eons led to the adaptation and acquisition of a high diversity of bacterial anti-phage strategies. The “pan-immune system” of bacteria comprising anti-phage defenses has been nicely reviewed by Bernheim and Sorek [9]. These strategies evolved by bacteria can be as simple as selecting for mutations in the phage receptor and preventing phage adsorption by modifying and adapting its lipopolysaccharide (LPS). Of greater concern are mutations in non-receptor host factors that can lead to phage cross-resistance, as was shown recently in *S. enterica* serovar Typhimurium [10]. Bacteria can also resort to elegant, more complex and specialized defense systems once the phage genome has been injected, such as Restriction-Modification (RM) [11], CRISPR-Cas [12], abortive infection (Abi) [13] comprising the recently described cyclic-oligonucleotide-based anti-phage signaling system (CBASS) [14], and bacteriophage Exclusion (BREX) [15,16,17] systems to name but a few. These systems are often passed on between bacteria via horizontal gene transfers. As we keep discovering new anti-phage defense systems, the selection of phages and the formulation of a cocktail should take bacterial defense against phage infection into consideration. Undoubtedly, co-evolution works both ways, and phages, therefore, evolved their own arsenal to counter cellular defenses. One can cite anti-RM or anti-CRISPR proteins [18] as well as the newly discovered anti-CBASS and anti-Pycsar proteins [19,20]. Given the diversity of anti-phage defense systems, a matching diversity of yet unknown anti-cellular defense functions is probably buried in the phage genomic “dark-matter” awaiting discovery. The phage “dark-matter” encompasses all those predicted genes in phage genomes and metaviromes with unknown functions [21]. It is important to collect as much information as possible on the phage genomes to select the best candidates when foreseeing phage therapy and to keep a balance between killing efficiency, susceptibility against cellular defenses, and potential antagonistic interactions between the therapeutical phages. A thorough genomic characterization of phage genomes, and especially functional annotation, are thus of crucial importance for the successful deployment of phage-based biocontrol strategies.

In this study, we sequenced and analyzed 10 new bacteriophage genomes infecting *Salmonella enterica* subsp. *enterica* serovar Typhimurium strain ATCC 14028S. These phages were isolated from wastewater and fresh water pounds in the Sevilla area in Spain. Most phages originate from a study by Olivenza et al. [22] aimed at developing epigenetic phage biosensors used to identify and select phages from different natural environments. Others belong to the personal collections of Dr. M. Ansaldi and Prof. J. Casadesús. To improve phage gene detection and annotation, we benchmarked five different gene callers for syntaxic annotation. We could ascribe a taxonomic affiliation for each phage at the genus level, thanks to genome-wide comparative analyses enabled by vContact2 [23] (based on the gene-sharing network) and VICTOR (based on genome-wide phylogeny) [24]. For functional annotation of the predicted ORFs, we combined complementary approaches based on remote homologies detection using Hidden Markov Model (HMM) protein profiles comparisons with PHROG [25], a newly published database of viral protein clusters, and the pdb70 database at the Protein Data Bank [26]. Thirdly, we specifically searched for anti-CRISPR proteins (Acr) using the integrated online platform AcrHub [27], relying on machine learning for Acr prediction. Acr are indeed valuable assets when foreseeing phage therapy. We further mined our annotated genomes for functions that may favor or, on the contrary, disfavor the selection of a phage for phage therapy.

The 10 newly isolated phage strains studied here all belong to the *Caudoviricetes* and are spread among four genera (*Kuttervirus*, *Chivirus*, *Jerseyvirus*, and *Ledergbervirus*) with a majority of six *Kuttervirus* phages isolated (*Ackermannviridae*). Eight of these ten strains are new species. One phage, Salmonella phage Salfasec13b (*Lederbergvirus*), was predicted as temperate, which is not a desired property for phage therapy. Among the most interesting features, we found phage-encoded proteins targeting the cellular defenses against phage infections, such as the Restriction-Modification, CRISPR-Cas, and Abortive infection systems. It is noteworthy that, Salmonella phage SeF3a (*Kuttervirus*) is likely to code for a novel anti-Abi system derived from the bacterial Phage Shock Protein A (PspA), making this phage an interesting candidate for phage therapy with two different anti-Abi systems, one found in all our *Kutterviruses* previously known to allow bacteriophage T4 to resist the Rex Abi system encoded in some prophages, and the second found in Salmonella phage SeF3a hypothesized in this study. We incidentally could propose functional annotations for 24 protein families in the PHROG database currently annotated “unknown function”. Altogether, this work highlights the need to design specific pipelines for phage genomic analyses to unravel the phage genomic “dark matter” and illustrates the importance of prior knowledge of functions encoded in phage genomes for an educated selection of phage candidates for therapeutic cocktails.

## 2. Results and Discussion

In this work, we used 14 phage isolates from wastewater and fresh water ponds in the Sevilla region in Spain, some of them detected and isolated by Olivenza et al. using an epigenetic phage biosensor [22].

### 2.1. Phage Genome de Novo Assembly

Genome de novo assembly with SPAdes led to the identification of a total of 18 meaningful contigs in the entire dataset with contig lengths between 40 and 160 kb, a range compatible with phage genome sizes (Appendix B, Table A1). Of note, SPAdes assembled two distinct contigs from Salfasec_11, Salfasec_13, Se_F3, and Se_F6 purified DNA, possibly representing two different phages in the same sample. This is in accordance with the TEM images (Figure 1), where we can visualize two different phage morphologies in purified phage suspensions for Salfasec_11 (two siphovirus with different tail lengths, red arrows), Se_F3 (one siphovirus with a small head and a long tail, and one myovirus, red and blue arrows, respectively), and Se_F6 (one siphovirus with a small head and a long tail and one myovirus, red and blue arrows, respectively) phage suspensions. In Salfasec_13 phage suspension, we could only identify one myovirus morphology (blue arrow) despite the assembly of two distinct genomes with very different sizes (157,296 and 59,161 bp).

When relevant, the two DNA sequences present in the same sample were distinguished by adding “a” and “b” to the sample isolate name. Experience in many labs dealing with phage isolation and sequencing demonstrated that the experimenters are not immune to sample cross-contaminations despite the important precautions taken, or that the same phage can be isolated several times from different samples. Resident prophage(s) can also be induced at a basal level in laboratory culture conditions or by the stress triggered by incoming phage infections. In such cases, one would expect more than one genome to be sequenced. As a reminder, our propagation strain *S. enterica* ATCC 14028S is lysogenic for at least three functional temperate bacteriophages, namely Gifsy-1, Gifsy-2, and Gifsy-3 [28], with a fourth one we predicted as complete and functional using PHASTER [29] in silico analysis. We chose to center our study on potential new phage species. We used VIRIDIC [30] to group the contigs summarized in Appendix B, Table A1 (species threshold similarity 95% and genus threshold similarity 70%). We thus could classify the 18 contigs into 4 different genera and 10 different species. Some species clusters contained 100% identical DNA sequences. For instance, all three contigs in species cluster 3 (Se_ML1, Salfasec_13a, and Se_F3a) are identical. We then selected 10 unique contigs representing the 10 different species clusters for downstream analyses (Table 1). The intergenomic similarity matrix, as well as the species and genus cluster, are shown in Appendix A.

Among these, only Salfasec_13b contig exhibits some small but significant DNA sequence identity with the propagation strain ATCC 14028S genome but only on 3 kb with 89.6% identity. This finding rules out that any of the sequences we obtained originate from resident ATCC 14028S prophages induced during phage propagation. Hereafter, these contigs will be referred to as Salmonella phage strains.

### 2.2. DNA Packaging Prediction

When DNA libraries were prepared from randomly fragmented DNA, as was the case for Se_F1, Se_F2, Se_F3a, Se_F6a, and Se_F6b, we used PhageTerm to predict putative genome termini and DNA packaging mode. Table 2 summarizes the prediction results.

According to PhageTerm predictions, Se_F1, Se_F3a, Se_F6a use headful (or *pac*) packaging. This is, for instance, the packaging mode of Salmonella phage P22 [31]. This mode of packaging implies that the first cut by the terminase occurs at a *pac* site on the DNA concatemer issued from the phage DNA replication cycle. The second cut occurs at a random site on the concatemer when the phage capsid is full (more than 1 genome unit length is packaged). The third cut occurs at a random site on the concatemer for the second capsid packaging, and so on. Accordingly, at the population level, we obtained circularly permutated genomes for Se_F1, Se_F3a, and Se_F6a. In contrast, Se_F2 and Se_F6b lay in the *cos* packaging group exemplified by Escherichia coli phage Lambda [32], although for Se_F6b, the low sequence coverage precludes valid termini prediction by PhageTerm. In this mode of packaging, the concatemer is cut at *cos* sites, and each phage capsid contains one identical genome unit length. We could not use PhageTerm for Se_AO1, Se_EM1, Se_EM2, Se_EM4, and Salfasec_13b due to the enzymatic DNA fragmentation method used for the preparation of the DNA libraries that does not allow recovering genome extremities (see Materials and Methods). Nevertheless, we could gain some information on their DNA packaging strategies. Paired-reads mapping with Bowtie2 [33] and mapping visualization with IGV [34] (data not shown) revealed circularly permutated sequences, suggesting that DNA packaging for these phages was probably through the headful mechanism.

### 2.3. Syntaxic Annotation of 10 Newly Isolated Salmonella Phage Genomes

It is common knowledge that predictions vary from one gene caller to another, resulting at the extreme in either missing “real” ORFs or predicting “false” ones, not to mention discrepancies in initiation codon predictions [35]. ORF prediction is rendered even more complex in bacteriophages due to their genome organization. Indeed, phage genomes are compact, with overlapping genes; in some cases, a gene can be entirely included in another gene, a situation called “overprinting”. This is the case, for example, for Rz-spanin-o in *E. coli* phage Lambda, which is entirely coded in the Rz-spanin-i gene [36]. Phage genomes are often organized in functional modules that can be acquired/modified by recombination with other viruses or mobile genetic elements or by recombination with their bacterial host and its eventual prophage(s); this mosaic organization resulting from many horizontal gene transfers makes it difficult for any gene caller to build uniform predictive models. The “phage language” is yet to be completely translated and is a current focus of many groups in the field, with great hopes put into deep learning techniques.

We, thus, compared the output of AMIGene, Glimmer, MetaGeneAnnotator, Phanotate, and Prodigal to identify the best algorithm (or combination of algorithms) to optimize the gene calling of our phage genomes. We used Se_AO1 as a case study before generalizing the method to the other nine phage genomes included in this study. In order to compare the different predictions, we built a table for each gene caller where an individual predicted ORF is identified by a stop codon and a coding strand (+ or −) without its predicted initiation codon. This strategy should avoid discrepancies in the predicted initiation codon between the different algorithms. We then compiled the five prediction sets in a single table for each phage with the following unique rule: a predicted ORF from a gene caller is deemed identical to an ORF predicted by another gene caller if they share the same stop codon on the same strand, regardless their respective predicted initiation codon that may be different between the two gene callers. For each phage, we indicated in the table for each predicted ORF whether it is predicted or not by each of the five gene callers. Results are listed in Appendix A, and gene callers’ comparison is illustrated in the Venn diagrams shown in Figure 2 for Se_AO1.

Figure 2a compares the results obtained with all five gene callers. Glimmer displayed the lowest (*n* = 144) and Phanotate the highest (*n* = 233) number of predicted ORFs. A total of *n* = 290 ORFs were predicted, 90 of which were shared by all algorithms (31%). According to this analysis, 69% of the predicted ORFs would depend on the specificity of the gene caller algorithms, which is very improbable. Glimmer predicted *n* = 144 ORFs, 32.6% of which were unique to this software; therefore, Glimmer seems ill-fitted to our purpose. Thus, in Figure 2b, we drew another Venn diagram omitting Glimmer predictions. On the *n* = 243 predicted ORFs, 76.5% were common to all algorithms (*n* = 186), which is an expected range if all predictors ran with a similar level of accuracy. Of note, Phanotate predicted 34 ORFs overlooked by the 3 other gene callers. This is not surprising as Phanotate has been purposely designed to identify ORFs in phage genomes.

Setting a cut-off limit for the gene size was not an easy task, as we did not want to keep false predictions or discard ORFs that do code for small proteins. One has to keep in mind that small ORFs in bacteriophages can encode small proteins enabling phages to evade host anti-phage systems. Among them, we found anti-CRISPR proteins (Acr), such as ACR3112-12 (52 aa) from Pseudomonas phage D3112 [37]; proteins preventing superinfection, such as the immunity protein Imm (83 aa) of bacteriophage T4 [38] or the lipoprotein Llp (77 aa) of bacteriophage T5 [39]; anti-RM proteins, such as Ocr (117 aa) of bacteriophage T7 [40]; the newly discovered anti-CBASS Acb (about 94 aa) found in an increasing number of phages of various phylogenetic background, such as Pseudomonas phages PaMx33, 35, 41, and 43, as well as the enterophageT4 [19,20].

As the identification of anti-CRISPR and other types of anti-host defenses proteins was one of our goals, we chose a conservative approach as a first approximation and kept all the ORFs predicted by AMIGene, MetaGeneAnnotator, Phanotate, and Prodigal for the 10 phage genomes studied here. When several possibilities for the initiation codon were available, we kept the longest version of the ORF. Across our 10 genomes, about 72% of ORFs were predicted by all 4 gene callers (slightly less for Salfasec_13b) and about 14.5% by Phanotate only (Appendix B, Table A2).

### 2.4. Taxonomic Affiliation

Due to the remarkable diversity of phage nucleotide sequences and the pervasive mosaicism of phage genomes, phage phylogeny does not follow traditional hierarchical phylogeny, as noted by Dion et al. [41], and it has always been a challenge to define universal gene markers to classify phages such as the 16S RNA commonly used for bacteria. The current taxonomy of viruses is evolving rapidly using genome-wide analyses, particularly the concept of shared orthologous protein families to infer evolutionary relationships between viral genomes [42,43]. One important advantage of using such methods to classify phages is that they do not rely on prior knowledge of protein functions.

We thus resorted to genome-wide analyses at the protein level for taxonomic classification with vContact2 and VICTOR. Figure 3a displays a representation generated by Cytoscape of the interaction network between *Caudoviricetes* bacteriophages generated by vContact2 and including our 10 newly isolated phages. In this representation, the strength of the connection linking two phages depends on the number of protein clusters they have in common.

vContact2 also defines viral clusters (VC) as regrouping phages that share a significant number of protein clusters. Each VC comprises viruses taxonomically classified at various levels (genus, sub-family, or family). Our 10 newly isolated phages fall into 4 different viral clusters whose members are colored in Figure 3b. Se_F2 and Se_F6b belong to the VC_20 (orange), comprising 22 viruses defined at the genus level (*Chivirus*), Se_AO1, Se_EM1, Se_EM2, Se_EM4, Se_F3a, and Se_F6a to the VC_34 (blue) comprising 63 viruses defined at the family level (*Ackermannviridae*), Se_F1 to the VC_75 (red) comprising 25 viruses defined at the sub-family level (*Guernseyvirinae*), and finally, Salfasec_13b to the VC_243 (green) comprising 16 viruses defined at the genus level (*Lederbergvirus*). The complete list of the VC generated by vContact2 is available in Appendix A.

For the phages that remained unclassified at the genus level after the previous analysis, we refined the classification with VICTOR using the “amino acid” option analysis to build phylogenetic trees for Se_F1 (Figure 4a) and Se_AO1, Se_EM1, Se_EM2, Se_EM4, Se_F3a, and Se_F6a (Figure 4b). In each case, a minimum of three genomes belonging to different families were used as outgroups.

Figure 4 unambiguously shows that Se_F1 belongs to the *Jerseyvirus* genus and that Se_AO1, Se_EM1, Se_EM2, Se_EM4, Se_F3a, and Se_F6a belong to the *Kuttervirus* genus.

To determine whether these 10 phage strains represent novel species or not, we first downloaded for each of the 4 genera (*Kuttervirus*, *Chivirus*, *Jerseyvirus*, and *Lederbergvirus*) all the species officially acknowledged in the latest version of the ICTV classification, then computed with VIRIDIC within each genus the intergenomic similarities table (Appendix A). With a species threshold of 95%, we, thus, could define 8 new species. Table 3 summarizes the final taxonomic affiliation of the 10 new phage strains studied in this work, 8 of which are new species. All phages in this table belong to the *Duplodnaviria* realm, *Heunggongvirae* kingdom, *Uroviricota* phylum, and *Caudoviricetes* class. The eight new phage species are noted in blue in Table 3 with a proposed name in the recommended binomial format. The proposed strain name is given in the first column.

This taxonomic classification supports the PhageTerm predictions for Salmonella phage SeF2 and SeF6b (*Chivirus* genus) as their predicted termini sequences are the same as the one experimentally determined for the *Chivirus* Salmonella phage Chi (5′-GGTGCGCAGAGC-3′) [44]. This gives credit to PhageTerm prediction for Salmonella phage SeF6b despite the low sequence coverage (Table 2).

### 2.5. Functional Annotation

To date, the most common and fastest method to tentatively ascribe a function to a newly predicted gene product is by comparing the ORF or gene product sequences with existing sequences deposited in various databases, such as the nr/nt databases at NCBI. This approach relies on the confidence one can have in previous annotations. It is sometimes tedious to trace back the reasons justifying current annotations. For proteins that finally end up with “unknown function” annotation, one has to keep a critical eye for several reasons: (i) the function can truly be a novel one (the phage “dark-matter”), (ii) the amino acid sequence is too divergent to detect homologies with proteins of known function using “simple” sequence alignment algorithms such as BlastP, or (iii) the gene product does not exist (an artifact due to the ORF prediction algorithms). Functional annotation can be improved thanks to methods developed to detect remote homologies, classify protein in orthologous groups, or fold primary sequences onto known secondary or tertiary structures. As noted by Chen et al. [45], these many different approaches can be combined to yield reliable functional predictions.

Here we chose to combine two complementary approaches for the functional annotation of our 10 phage genomes.

Comparison of the predicted gene products with the newly published PHROG database comprising protein sequences exclusively from viruses infecting bacteria and archaea as well as their prophages. In PHROG, viral proteins are clustered in protein orthologs families called “phrog” built on remote homology detection and functionally annotated.Comparison of the predicted gene products with protein structures from the pdb70 database at the Protein Data Bank.

We, thus, first compared all predicted gene products for each phage genome with the PHROG database. Appendix A recapitulates the best hit in PHROG for each predicted ORF for each genome, as well as the comparison metrics. We then used the comparison metrics and the genomic environment to perform a manual curation of the predicted ORFs. In total, 59 ORFs were thus manually curated (highlighted in red in Appendix A), ranging from 1 curated ORF per genome (Salmonella phage SeF2 and SeF6b) to 10 curated ORFs per genome (Salmonella phage SeAO1). Of these, 89.8% of curated ORFs (53 ORFs) were predicted by Phanotate, suggesting that this gene caller designed for phage genomes tends to overpredict ORFs. For the remaining ORFs annotated “unknown function” with a weak affiliation to a phrog family, we replace the phrog number with “singleton”, meaning that this ORF does not have (yet) any ortholog in the PHROG database. This can be either a false positive ORF or a truly novel function. 

For gene products still annotated with “unknown function” after comparison with the PHROG database, we tried to improve the annotation by comparison with the pdb70 database of the Protein Data Bank. The best hits for each gene product of each genome are listed in Appendix A. After manual inspection of the results and the comparison metrics, we could propose a functional annotation for 24 phrog families annotated “unknown function”. These propositions are summarized in Table 4.

The proposed annotations cover 1120 protein sequences included in the PHROG database, 352 of which are structural proteins, and 196 participate in gene regulation according to our proposed annotations. Of note, phrog_6201 is predicted as a CI-like repressor, one of the usual markers of the temperate bacteriophages. This phrog contains 20 protein sequences from 20 distinct phages that are, thus, likely to be temperate phages. Of note, Salmonella phage Salfasec13b Gp_057 protein belongs to the phrog_6201, hinting at a temperate lifestyle for this bacteriophage. This example illustrates that one can gain valuable functional insights into an entire set of bacteriophages by improving the functional annotation of a single phrog. Table 5 summarizes the final syntaxic and functional annotation for each phage. Detailed functional annotations for each genome can be found in Appendix A.

The files for submission were built using the PHROG annotation for each ORF; when pertinent, the predicted functions derived from the PDB comparisons were added together with its corresponding PDB entry. As the PHROG database is not yet recognized by the International Nucleotide Sequence Database Collaboration (INSDC), it was not possible to use the “db_xref” flag for the PHROG annotations in the GenBank or ENA file. As a consequence, we reported the phrog number and the corresponding functional annotation for each gene as a “note” in the GenBank or ENA file. The 10 bacteriophage genomes were submitted to GenBank (Salmonella phage SeF1, SeF2, SeF3a, SeF6a, SeF6b, Salfasec13b, BioProject PRJNA767534) or ENA (SeAO1, SeEM1, SeEM2, SeEM4, BioProject PRJEB37792) under the accession numbers listed in Table 5.

### 2.6. Phage Lifestyle

Consensually, the development of phage therapy strategies relies on strictly virulent bacteriophages to formulate cocktails. The exclusion of temperate phages aims to avoid lysogenization that (1) would not immediately kill the targeted cell, (2) could enable the acquisition of virulent genes or other genes, increasing the fitness of the lysogens. In a somehow iconoclast view, Monteiro et al. [46] recently argued that temperate phages could be interesting as they are abundantly present in bacterial genomes, ready to be used and eventually engineered, although current regulations towards genetically modified organisms do not yet favor such approaches. Hence, the prediction of phage lifestyle prior to therapeutic cocktail formulation still remains a prerequisite. We scanned each phage proteome for predicted functions related to the bacteriophage lifestyles (virulent or temperate). When no integrase together with other functions generally associated with temperate phages, such as a Recombination Directionality Factor (RDF, also called excisionase) or CI-like repressors, were found, we hypothesized a “virulent” lifestyle. Such was the case for all phages except Salmonella phage Salfasec13b.

We believe that Salmonella phage Salfasec13b (*Lederbergvirus* genus) is a temperate phage as it contains many functions associated with temperate phages, such as an integrase of the tyrosine recombinase family (Gp_026, phrog_216), a CIII anti-termination protein (Gp_041, phrog_550), a CI-like repressor (Gp_057, phrog_6201), and a CII-like regulator (Gp_060, phrog_725). However, we could not detect a Recombination Directionality Factor (RDF or excisionase) required for prophage excision. RDFs are difficult to predict as they are small proteins sharing very few sequence homologies (66 aa for TorI, the RDF for the defective prophage KplEI in *E. coli* K12 [47,48]). It is, thus, probable that Salmonella phage Salfasec13b RDF is located among the small ORFs of unknown function. RDF genes are commonly found in the vicinity of their cognate integrase genes. Of note, in Salmonella phage Salfasec13b, the integrase gene (*gp_026*) is surrounded by predicted ORFs encoding a small protein of unknown function (Gp_027, 56 aa) compatible with the size of an RDF.

Salmonella phage SeF1 (*Jerseyvirus* genus) is an interesting case as it contains a predicted RDF (Gp_053, 76 aa, phrog_66) whose modeled structure we found matches very well with TorI (PDB 1Z4H), the RDF of the defective prophage KplE1 mentioned above. Nevertheless, no ORF coding for an integrase could be predicted, and other key functions associated with temperate phages are missing. The same findings were made for the closest annotated sequence homolog found with Blastn in the nt database, i.e., Salmonella phage S101 (Table 2). The PHROG database includes 22 *Jerseyvirus* genomes, 20 of them coding for RDF belonging to the phrog_66, making it a marker of this genus, although *Jerseyvirus* seem to be strictly virulent phages. Salmonella phage SeF1 and its relatives might have been temperate phages that have lost their integrase gene, or the RDF encoded in their genome has been acquired by horizontal gene transfer and may serve to induce resident prophage(s) in the host. The latter hypothesis is worth investigating, as induction of resident prophages is likely to ensure efficient lysis of the host.

### 2.7. Genetic Mining of 10 Newly Isolated Salmonella Phage Genomes in the Light of Phage Therapy

The primary aim of our isolation of new *Salmonella*-targeting phages is the development of phage therapy against *Salmonella* species, and we were particularly interested in two major kinds of protein functions.

Those that could give a selective advantage for host infection, successful virion productions, and cell lysis (mostly features allowing evasion from cellular defenses).Those that could interfere with the formulation of a bacteriophage cocktail, such as functions allowing lysogeny or super immunity against other phage infections.

We first reviewed these two categories of functions by the bacteriophage genus before focusing on the prediction of anti-CRISPR proteins, which belong to the first category of proteins likely to confer a selective advantage.

#### 2.7.1. *Jerseyvirus*: Salmonella Phage SeF1

We identified a DarB-like Type-I RM anti-restriction protein (Gp_039, phrog_10089). In bacteriophage P1 infecting *E. coli*, several anti-restriction proteins, including DarB, are packaged during virion assembly and released upon infection into the host cytoplasm, ready to act against host RM systems [49]. DarB alone is required for protection against type-I EcoB and EcoK restriction. The Gp_039 annotation (DarB-like anti-restriction) is sound because this protein belongs to phrog_10089, predicted to be highly similar (probability 100%, Evalue 3E-210) to phrog_1685, which does contain Bacteriophage P1 DarB. It, thus, seems that Salmonella phage SeF1 encodes at least a DarB-like anti-restriction protein. Of note, *Salmonella enterica* Typhimurium codes for SB, an enzyme similar to *E. coli* EcoA, EcoB, and EcoK type-I restriction enzymes [50]. The Salmonella phage SeF1 genome contains six restriction sites recognized by SB.

Gp_023 is predicted as an anti-repressor of the Ant type (phrog_130). In Salmonella prophages such as Gifsy-1 and Gifsy-2, prophage induction is repressed by the binding of the repressor Rep on the upstream promoter genes, allowing prophage induction [51]. The production of Ant derepresses these genes by titrating the Rep protein, thus triggering prophage(s) induction and eventually cell lysis. Gp_023 expression is then susceptible to inducing resident prophage(s) in the infected host and may then help host takeover. Together with the RDF mentioned previously, this is the second protein indicative of a temperate lifestyle, although Salmonella phage SeF1 lacks the other functions required for lysogenization.

Gp_027 is predicted as an immunity-to-superinfection protein (phrog_1039). In *E. coli* phage T4, the immunity protein Imm blocks the injection of T4 DNA, preventing superinfection by the same phage or another phage targeting the same bacterium and using the same infection mechanisms. Bacteriophage T4 Imm belongs to the same phrog as Gp_027, strengthening the proposed annotation for Gp_027. The production of an Imm-like protein in an infected cell will alter co-infection by another virion, either from the same strain or from a bacteriophage that resorts to the same DNA injection mechanism. 

Salmonella phage SeF1 seems to harbor an extensive arsenal for host lysis with two Rz-like spanins (Gp_009 and Gp_010, phrog_2536 and phrog_5672, respectively), one endopeptidase predicted from structural comparison with the pdb70 (Gp_034, phrog_12508 currently annotated “unknown function” in the PHROG database), two holins (Gp_063 and Gp_064, phrog_2359 and phrog_2508, respectively) and one endolysin (Gp_065, phrog_33083).

#### 2.7.2. *Chivirus*: Salmonella Phage SeF2 and SeF6b

Salmonella phage Chi and other Chi-like viruses are flagellotropic phages requiring a motile flagellum as its primary receptor; it is then very likely that Salmonella phage SeF2 and SeF6b share the same requirement. Both Salmonella phage SeF2 and SeF6b phages contain 3 gene products with predicted carbohydrate-binding domains that we predicted from structural comparison with the pdb70 (phrog_2448 currently annotated “unknown function” in the PHROG database) encoded on the same strand downstream of the tail genes module, suggesting that these proteins may be part of the tail and the host recognition apparatus. These genes encoding carbohydrate-binding domains are shared among 36 other *Chivirus* present in the PHROG database. Sugars are components of the LPS, and it has also been shown that *S. enterica* Typhimurium flagella contain at least 16 different sugars [52]. The flagellotropic Salmonella phage Chi and its relatives may use these domains to recognize and bind the bacterial flagellum and its sugar moieties. It has been reported that Salmonella phage Chi can infect both *Salmonella* spp. and *E. coli* hosts and thus has been proposed as a good candidate for the development of phage-based applications (pathogens detection, remediation, phage therapy) [53]. Inclusion in a therapeutical cocktail of bacteriophages such as *Chivirus* that may target receptors other than the LPS would be a good option to avoid cross-resistance.

Both phages encode a predicted DNA methyltransferase (phrog_111) homologous to the DNA N-6-adenine methyltransferase according to PFAM prediction (PF05869) attached to this phrog. This enzyme may serve to modify phage DNA during replication, allowing the phages to escape host restriction–modification systems.

#### 2.7.3. *Lederbergvirus*: Salmonella Phage Salfasec13b

As discussed above, Salmonella phage Salfasec13b is a temperate phage and a *Lederbergvirus* (as the archetypal Salmonella phage P22). This phage genome contains all the genes found in P22 that are important to promote lysogeny or confer a selective advantage to the host including *gtrA* (Gp_022, phrog_2224), *gtrB* (Gp_021, phrog_34335), and *gtrC* (Gp_019, phrog_4829). Together, the GtrABC complex modifies the host O-antigen at the host cell surface [54]. Changing the LPS prevents infection by other phages using it as a primary receptor. In the light of phage therapy, this is not the desired outcome.

#### 2.7.4. *Kuttervirus*: Salmonella Phage SeAO1, SeEM1, SeEM2, SeEM4, SeF3a, SeF6a

We first pinpoint protein functions that seem to be a hallmark of the *Kuttervirus* genus as they are found both in our 6 *Kuttervirus* phage genomes and in almost all the 15 *Kuttervirus* genomes included in the PHROG database.

An interesting feature is the gene product belonging to phrog_1510 (superinfection exclusion) that includes the Gp17 protein of Salmonella phage P22. In the classical literature, Gp17 is described as a protein necessary for P22 to counteract a superinfection exclusion system encoded in the Fels-2 prophage found in many *S. enterica* Typhimurium strains [55]. A sensible hypothesis is that *Kuttervirus* phages have somehow acquired a P22 *gp17-like* gene that allows them to successfully infect and propagate in lysogenic *Salmonella* strains harboring Fels-2-like superinfection exclusion systems. This is an interesting feature for a phage cocktail, as *Salmonella* spp. harbor many diverse prophages [56].

Another hallmark is phrog_600 (unknown function), which we could predict as a lysozyme-like protein by comparison with the pdb70. This adds to the predicted Rz-spanin and endolysin in the arsenal of proteins, ensuring host lysis.

The RIIA (phrog_3661, RIIA lysis inhibition) and RIIB (phrog_3803, RIIB lysis inhibition) are also encoded in most *Kuttervirus* phages. The phage proteins RIIA and RIIB enable bacteriophage T4 to resist the Rex restriction system encoded by *E. coli* Lambda lysogens. Rex is an abortive infection defense system and relies on the action of two proteins, RexA and RexB [57]. However, the molecular mechanisms behind Rex still remain elusive. RIIA and RIIB could confer an advantage to *Kuttervirus* phages when infecting lysogens encoding a Rex-like system.

We were intrigued by the presence in all our six *Kuttervirus* genomes of a predicted second terminase small subunit (phrog_800) situated far away from the traditional pair of consecutive genes encoding the small and the large terminase subunits. To the best of our knowledge, this is unheard of in phage genomes. A closer inspection of phrog_800 revealed a probable misannotation. Indeed, by checking the list of PDB hits (Appendix A), we found that phrog_800 proteins are predicted to be deoxynucleoside kinase proteins. In our genomes, the phrog_800-encoding gene is followed by genes encoding a thymidiate synthase, a deoxynucleoside monophosphate kinase, and finally, a nucleoside triphosphate pyrophosphohydrolase. This genomic environment strongly suggests a gene cluster involved in DNA metabolism and strengthens our proposed new annotation for phrog_800 as a deoxynucleoside_kinase.

Finally, an interesting feature was found only in Salmonella phage SeF3a. Our comparison with the pdb70 strongly suggests that Gp_125 (phrog_1203, unknown function) is homologous to PspA, the Phage Shock Protein A. Four of the fifteen *Kuttervirus* phage genomes included in the PHROG database harbor such a gene. *pspA* belongs to the *psp* operon found in some bacterial genomes, noticeably in *E. coli* and *Salmonella* species. This system prevents the dissipation of the proton-motive force (PMF) triggered by various stresses, including the infection by filamentous phages or the mislocalization of secretins [58]. PspA plays a role in the regulation of the phage-shock-response *psp* operon but also associates with the inner membrane to prevent the loss of PMF, although the molecular mechanisms still remain unclear. Several Abi systems, such as Rex in Lambda lysogens mentioned above or AbiZ in *Lactococcus lactis* [13], lead to cell death or cellular growth arrest through membrane permeabilization and dissipation of the PMF or loss of ATP. A reasonable hypothesis could be that the phage-encoded PspA acts as a novel anti-Abi system, preventing cell suicide. This hypothesis is worth further investigation. This feature singles out Salmonella phage SeF3a from the other *Kuttervirus* phages isolated in this study, with a putative additional anti-host defense to its arsenal.

#### 2.7.5. Prediction of Anti-CRISPR Proteins (Acr)

*Salmonella enterica* species possess a type I-E CRISPR-Cas system with a *cas* operon and two CRISPR arrays [59]. This adaptative immune defense system may interfere with phage infection; hence, searching for phage-encoded anti-CRISPR (Acr) proteins is interesting in the light of phage therapy. We performed Acr encoding genes prediction with AcrHub. Among all the Acr predicted in our 10 phage genomes, we retained the most probable predictions, summarized in Table 6.

In all phage genomes studied here, we could retain at least one Acr encoding gene prediction. Salmonella phage SeF1 harbors the highest number of predicted Acr (*n* = 6). These predictions are generally consistent for bacteriophages belonging to the same genus. For *Kuttervirus* genomes, phrog_4097 is found in all six genomes studied, and phrog_519 in five out of six genomes. For *Chivirus* genomes, Salmonella phage SeF2 and SeF6b genomes code for the same three predicted Acr. 

Interestingly, the transposable Pseudomonas aeruginosa Mu-like phage genomes contain an *acr* locus comprising two to three genes located among the genes involved in the phage head assembly [37]. In these phages, the *acr* locus is bracketed by genes involved in phage head morphogenesis, usually upstream of a protease/scaffold gene. The lower panel in Figure 5 describes the genetic context of AcrIE3, a type-IE Acr protein from P. aeruginosa phage DMS3 that has been functionally investigated [60]. In our six *Kuttervirus* phages, we found one small protein (56 aa) belonging to phrog_4097 (unknown function) predicted with good scores by both PaCRISPR and AcRanker as a type II-A anti-CRISPR protein. The gene encoding this protein in the *Kuttervirus* genomes is situated between the portal protein and head scaffolding protein-encoding genes, reminiscent of the genetic context of AcrIE3 (Figure 5, upper panel).

One can also find a similar gene coding for a phrog_4097 protein in the same genomic context in 14 of the 15 *Kuttervirus* genomes found in the PHROG database, making this predicted Acr another hallmark of Kuttervirus genomes. This predicted Acr is, thus, a serious candidate for anti-CRISPR functional studies and an interesting feature for therapeutical cocktails.

Salmonella phage SeF1 Gp_057 exhibits high scores for both PaCRISPR and AcRanker predictions. Although we could not link gp_057 genomic context to known contexts of experimentally validated Acr, we believe Gp_057 is also a serious candidate for anti-CRISPR functional studies.

## 3. Materials and Methods

### 3.1. Phage Lysates and Bacterial Strain

Se_AO1, Se_EM1, Se_EM2, Se_EM3, Se_EM4, Se_F1, Se_F2, Se_F3, Se_F6, Salfasec_9, Salfasec_10, Salfasec_11, and Salfasec_13 culture lysates were kindly provided by Dr. Olivenza (Departamento de Genética, Facultad de Biologia, Universidad de Sevilla, 41012 Sevilla, Spain). All phages were isolated in the Sevilla area (Spain) from wastewater or freshwater pounds. When required, *Salmonella enterica* subsp. *enterica* serovar Typhimurium ATCC 14028S (ATCC 14028S) was used to propagate and titrate bacteriophages. Cells were grown aerobically in either Lysogeny Broth (LB) at 37 °C under shaking (180 rpm) for liquid cultures or on LB-1% agar plates incubated at 37 °C.

### 3.2. Phage Propagation, Purification, and Titration

To propagate phages from culture lysates, 200 mL Erlenmeyer flasks containing 25 mL of LB were inoculated with an overnight (ON) culture of ATCC 14028S at an optical density measured at 600 nm (OD_600nm_) of 0.04. Cells were grown until OD_600nm_ reached 0.4, then inoculated with 100 µL of 0.22 µm-filtered and chloroform-treated culture lysate. The culture was further incubated for about 4 h. Cells were lysed by the addition of 10% chloroform (vortexed for 15 s and incubated for 5 min at room temperature, repeated twice). The aqueous supernatant (phage lysate) was recovered after centrifugation (7500× *g*, 15 min) to pellet the cell debris and separate the aqueous phase from the solvent. The phage lysate was then 0.22 µm-filtered and stored at 4 °C. Phage titration by double agar overlay plaque assay was performed according to Kropinski et al. [61]. Phage titers are expressed in Plaque Forming Unit per mL (PFU/mL). For further phage purification and concentration, all centrifugation steps were done at 4 °C, 20,800× *g* for 1 h, and ice-cold buffers were used. A total of 4 mL of phage lysate (titer between 1.10^10^ and 3.10^11^ PFU/mL) were centrifuged, and the pellet was resuspended in 2 mL of 0.22 µm-filtered Phage Buffer (PB: 10 mM Tris-HCl pH 7.5, 100 mM NaCl, 25 mM MgCl_2_, 1 mM CaCl_2_). Phages were pelleted once again and resuspended in 200 µL of PB. Purified phage suspensions were stored at 4 °C.

### 3.3. Phage Morphological Characterization by TEM

For Transmission Electronic Microscopy (TEM) exploration, the final phage pellet resuspension was performed in 2 mL of 0.22 µm-filtered TEM buffer (0.1 M ammonium acetate pH 7). Phages were pelleted once again and resuspended in 20 µL of TEM buffer. Formvar/carbon-coated copper grids were prepared at the Institut de la Méditerranée (IMM) Microscopy facility. A total of 5 µL of the phage solution was pipetted onto the grid surface and allowed to sediment for 3 min at RT. Excess liquid was then blotted and grids negatively stained according to Ackermann’s protocol with 2% uranyl acetate. Observations were made on an FEI Tecnai G2 20 TWIN (200KV), laB6, Gatan Oneview 4k × 4k CMOS transmission electron microscope. Images were visualized using FIJI [62].

### 3.4. Phage DNA Purification

To remove non-encapsidated nucleic acids, 20 µL of DNAse I (6 mg/mL, EUROMEDEX, Souffelweyersheim, France), 2 µL of RNAse A (4 mg/mL, Promega), and 4 µL of DpnI restriction enzyme (20,000 U/mL, New England Biolabs, Ipswich, MA, USA) were added to 200 µL of purified phages in PB. The sample was then incubated for 1 h at 37 °C. DNAse I was inactivated by incubating the sample for 20 min at 65 °C under shaking. Phage DNA was released from the capsid by adding 20 µL of 10% SDS and 20 µL of proteinase K (50 µg/mL, EPICENTRE, LGC, Teddington, UK) and incubating the sample for 1 h at 56 °C. The sample volume was completed with PB buffer q.s. 600 µL. Phage DNA was extracted by adding 600 µL phenol/chloroform/isoamyl alcohol (25:24:1, Sigma-Aldrich, St. Louis, MO, USA) and then vortexing the sample for 30 s. The upper aqueous phase containing the nucleic acids was recovered after centrifugation (10,000× *g*, 10 min, 4 °C). The extraction was repeated twice. The aqueous phase-containing DNA was then ethanol-precipitated. The DNA pellet was finally resuspended in 20 µL DNAse/RNase-free water and stored at −20 °C. Phage dsDNA was quantified with a Qubit™ fluorometer in combination with the Qubit™ dsDNA HS Assay kit (Invitrogen).

Phage genomic DNA purified from Se_F1, Se_F2, Se_F3, Se_F6, and Salfasec_11 phage lysates was mechanically fragmented in 50 µL microtubes using a Covaris M220 sonifier with the following parameters: peak power 75W, duty factor 10%, Cycles/Burst 200.

Phage genomic DNA purified from Se_AO1, Se_ML1, Se_EM1, Se_EM2, Se_EM3, Se_EM4, Salfasec_9, Salfasec_10, and Salfasec_13 phage lysates was enzymatically fragmented by the tagmentase technology from the NEXTERA XT kit (Illumina^®^, San Diego, CA, USA) according to the manufacturer’s protocol.

DNA libraries for high throughput sequencing were prepared from the fragmented DNA at the IMM Transcriptomic and Genomic facility with the NEBNext^®^ Ultra™ II DNA Library Prep kit for Illumina^®^ (New England Biolabs, Ipswich, MA, USA) for Se_F1, Se_F2, Se_F3, Se_F6, Salfasec_11, and with the NEXTERA XT kit (Illumina^®^, San Diego, CA, USA) for Se_AO1, Se_ML1, Se_EM1, Se_EM2, Se_EM3, Se_EM4 according to the manufacturer’s protocols.

Salfasec_9, Salfasec_10, and Salfasec_13 DNA libraries preparation with the NEXTERA XT kit (Illumina^®^, San Diego, CA, USA) was subcontracted to AllGenetics (A Coruña, Spain).

### 3.5. High Throughput DNA Sequencing

Prior to sequencing, Se_AO1, Se_EM1, Se_EM2, Se_EM3, Se_EM4, Se_ML1, Se_F1, Se_F2, Se_F3, Se_F6, and Salfasec_11 DNA libraries were quantified with a Qubit™ dsDNA HS Assay kit (Invitrogen, Waltham, MA, USA) and their size distribution profiles recorded with the TapeStation 4200 System (Agilent) in combination with the D5000 DNA ScreenTape System (Agilent, Santa Clara, CA, USA). Libraries were then diluted at 4 nM in the appropriate buffer. Paired-end (2 × 150 bp) DNA sequencing was performed on the MiSeq sequencer (Illumina^®^, San Diego, CA, USA) hosted at the IMM Transcriptomic and Genomic facility with a MiSeq v2 (300-cycles) flow cell (Illumina^®^, San Diego, CA, USA) according to the manufacturer’s protocol.

Salfasec_9, Salfasec_10, and Salfasec_13 DNA libraries sequencing was subcontracted to AllGenetics (Coruña, Spain).

Raw sequencing reads (FASTQ files trimmed from their Illumina adaptors) for each BioSample were submitted to the NCBI Sequence Read Archive under the BioProject accession number PRJNA767534. Raw sequencing reads quality was then improved with Trimmomatic [63] using the following parameters: SLIDINGWINDOW:4:25 MINLEN:75 for Se_AO1, Se_ML1, Se_EM1, Se_EM2, Se_EM3, and Se_EM4 or ILLUMINACLIP TruSe3-SE.fasta SLIDINGWINDOW:4:28 MINLEN:75 for all other phages. Only trimmed paired-reads were used for de novo genome assembly.

### 3.6. Genome de Novo Assembly

Genome de novo assembly was performed with SPAdes 3.14.1 with default parameters. Table A1 (Appendix B) summarizes the size of the 18 meaningful contig(s) obtained for the 14 BioSamples together with their mean genome coverage. By “meaningful”, we mean contigs with significant sequence coverage and length compatible with a phage genome (50 kb average size for dsDNA phages) and contigs that are not contaminants. Contaminants are usually small contigs with low sequencing coverage and whose DNA sequences match with unrelated organisms (Blastn analyses). For most contigs, we could identify identical 77 bp-DNA sequences at each contig extremity due to the SPAdes assembly algorithm. We observed this with circularly permuted genomes. We later removed the right-most copy to obtain the final contigs we considered as circular for downstream syntaxic annotation, although we are well aware that the encapsidated biological DNA molecule is linear. Indeed, we found out that some genes were spanning the contig extremities. Such was the case for Se_F2 and Se_F6b genomes.

### 3.7. Genomes Clustering

We used VIRIDIC (Available online: http://rhea.icbm.uni-oldenburg.de/VIRIDIC/) (accessed on 27 July 2022) to cluster the 18 contigs with 95% and 70% for the species and genus threshold, respectively, and the following BLASTN parameters: ‘-word_size 7-reward 2-penalty-3-gapopen 5-gapextend 2’. Genomes from the *Kuttervirus*, *Chivirus*, *Jerseyvirus,* and *Lederbergvirus* sanctioned by the ICTV (Master Species List 2021_v2) were downloaded from NCBI.

### 3.8. Prediction of DNA Packaging

We used PhageTerm [64] to predict putative phage genome termini as well as the DNA packaging mode. PhageTerm is based on the statistical analysis of the sequence coverage after mapping the short sequencing reads onto the assembled contig. For a statistically sound result, PhageTerm requires a minimum sequence coverage of 50. Obviously, PhageTerm requires that the genome extremities are sequenced, and this depends on the technology used to construct the DNA libraries for sequencing. Genome extremities can be recovered when the phage DNA has been mechanically fragmented, as described above. This was the case for Se_F1, Se_F2, Se_F6, and Salfasec_11. When the tagmentation has been used to construct the libraries (Nextera XT kit from Illumina^®^, San Diego, CA, USA), one cannot recover the extremities of a linear dsDNA, which precludes termini identification by PhageTerm when DNA packaging is initiated by the terminase at fixed positions (*cos* or *pac* sites) and ends at a fixed position (*cos* site). Tagmentation was used for Se_AO1, Se_ML1, Se_EM1, Se_EM2, Se_EM3, Se_EM4, Salfasec_9, Salfasec_10 and Salfasec_13.

### 3.9. Syntaxic Annotation

For tRNA prediction, we used ARAGORN [65]. For Open Reading Frames (ORF) prediction, we tested five different gene callers. Indeed, many gene callers are freely available to predict ORFs from nucleic acid sequences, but they rely on different algorithms, leading to variable predictions. We selected four of them optimized for bacteria (AMIGene, Glimmer, MetaGeneAnnotator, and Prodigal) and one optimized for phage genomes (Phanotate) [66,67,68,69,70].

Locus tag names were built with the same convention for each phage: an alphanumeric string defining the phage name coupled with an alphanumeric string defining the gene product numbering. For instance, Se_AO1_GP_001 refers to the first ORF predicted for *Salmonella* phage SeAO1. When present, tRNAs were labeled following the same convention: Se_AO1_tRNA1 refers to the first tRNA predicted for the *Salmonella* phage SeAO1 genome. To ease further genomic comparisons, we decided to start the ORFs numbering with the terminase small subunit-encoding ORF as *gp_001*. All DNA sequences were flipped and/or rotated accordingly.

### 3.10. Taxonomic Classification

For taxonomic classification, we used vContact2 0.9.19 (with the ProkaryoticViralRefSeq201 database accessed on 27 September 2021) and VICTOR (Virus Classification and Tree Building Online Resource with the amino acid option selected, Available online: https://ggdc.dsmz.de/victor.php) (accessed on 15 November 2021) Network visualization was done with Cytoscape [71].

### 3.11. Detection of Protein Sequence Remote Homologies

We generated Hidden Markov Model (HMM) profiles for each predicted protein sequence of each phage with HHblits from the UniRef30_2020_06 database in order to detect sequence remote homologies [72,73].

### 3.12. PHROG-Based Functional Annotation

For functional annotation, we used the newly published PHROG database (accessed on 27 September 2021) dedicated to viral proteins. In this database, viral proteins are distributed among families called “phrog”; each phrog represents a cluster of viral proteins orthologs built using remote homology detection by HMM profile–profile comparisons. At the date of its publication, the PHROG database contains 17,473 (pro)viruses of prokaryotes or archaea for a total of 938,864 proteins. A total of 38,880 clusters were defined, and manual inspection led to the annotation of 5108 phrog clusters, representing 50.6% of the total protein dataset. Following the procedure available on the PHROG website (https://phrogs.lmge.uca.fr/READMORE.php), we used HH-suite [74] to compare each predicted gene product from the phages studied here with the PHROG database and ascribe a phrog and its annotation whenever possible. When several hits were found, we kept the best one for the considered protein. The corresponding phrog number and its functional annotation were then transferred to our newly predicted protein. When no hit with the PHROG database was found, or the affiliation to an existing phrog was too distant (probability < 80%, Evalue > 1 × 10^−4^), the ORF was annotated as a singleton of unknown function according to the PHROG guidelines. The entire list of the best hit for each ORF for each phage before manual curation is supplied in Appendix A.

### 3.13. PDB-Based Functional Annotation

In order to improve the PHROG-based functional annotation for protein still annotated “unknown function”, we performed with HHblits a comparison of each predicted protein sequence HMM profile with the Protein Data Bank pdb70 database (accessed on 28 September 2021) to detect homologies with known protein structures. We kept the first hit for each protein sequence for further consideration when the probability was greater than 80% and the Evalue less than 1 × 10^−3^. When these criteria were met, we also manually checked the prediction coverage to avoid transferring a PDB annotation based on a predicted structural similarity between a small portion (a domain or a sub-domain) of the PDB hit and the phage protein sequences. After all these considerations, we manually checked for phage ORF ascribed to a phrog of unknown function if a reliable structural prediction from the pdb70 could be used instead. If so, a new annotation inferred from the structural prediction was transferred to the corresponding phrog of unknown function (see Table 4). A new annotation for this phrog cluster will be proposed to the PHROG database through a dedicated form on its website (https://phrogs.lmge.uca.fr/suggestions.php).

### 3.14. ORFs Manual Curation

Phage genes sometimes overlap or may even be entirely included in one another. Nevertheless, in some cases, these overlaps can reveal misprediction from the syntaxic annotation. Thus, in an attempt to reduce false positive predictions, we manually scanned each genome to detect highly overlapping ORFs. The most common situation we encountered was two highly overlapping ORFs, with one ORF coded on one strand with a strong affiliation to a phrog family while the ORF encoded on the opposite strand had a weak affiliation to a phrog family (probability < 70%, Evalue > 1 × 10^−4^). If the latter did not show any strong homology with a known structure from the PDB (probability > 80%, Evalue < 1 × 10^−3^), we decided to discard the ORF. As an example of manual curation, for Salmonella phage SeAO1, *gp_014* and *gp_016* almost entirely overlap with *gp_015*, all 3 ORFs predicted on the same strand. According to the phrog affiliation metrics generated by HHblits (see Appendix A), Gp_015 is strongly affiliated to phrog_519 (Prob. 99.5%, Evalue 1.4 × 10^−18^, 261 proteins sequences in this phrog family), whereas Gp_014 is weakly affiliated to phrog_32723 (Prob. 55%, Evalue 1.1, 2 proteins sequences in this phrog family) and Gp_016 is weakly affiliated to phrog_8445 (Prob. 23.5%, Evalue 7.4, 13 proteins sequences in this phrog family). In this example, we discarded *gp_014* and *gp_016*. In other examples, such as *gp_212*, *gp_213,* and *gp_214*, it was not possible to decide which ORF was significant based on the phrog affiliation metrics; all three ORFs, although overlapping, code for proteins strongly affiliated to a phrog family. In this case, we chose to keep all three ORFs. The last example of manual curation is *gp_222*. Based on phrog affiliation metrics, we classified Gp_222 as a “singleton” of unknown function. However, *gp_222* completely overlaps with the reliably predicted tRNA6-Val on the same strand; we, thus, decided to discard *gp_222*. Table 5 summarizes the final gene content predicted for each phage after manual curation. This procedure was carried out for every genome included in this study.

### 3.15. Prediction of Anti-CRISPR Proteins (Acr)

Until very recently, searching for Acr was tedious, essentially relying on “guilt by association” with anti-CRISPR-associated (Aca) proteins containing “helix-turn-helix” motives. However, not all Acr are associated with an Aca. Thanks to the new approaches based on machine learning, it is now possible to predict Acr found in undescribed genomic environments. AcrHub is a platform that incorporates state-of-the-art Acr predictors and three analytical modules (similarity analysis, phylogenetic analysis, and homology network analysis) [27]. The AcrHub database contains 339 experimentally validated, and 71,728 predicted Acr proteins. Validated Acr are mostly short proteins (89% between 60 and 160 aa). We ran our phage proteomes in AcrHub on two different Acr prediction algorithms (PaCRISPR [75] and AcRanker [76]). Meaningful score thresholds are over 0.5 for PaCRISPR and over −5 for AcRanker. We then used the AcrHub “Similarity analysis” module to relate our predicted Acr with experimentally validated Acr proteins in the AcrHub database. We excluded overlong proteins (>200 aa) and those whose query cover was below 40%. When pertinent, we compared the genomic surrounding of our predicted *acr* gene with an experimentally validated *acr* gene product.

## 4. Conclusions

In this study, we present a genomic description of 10 phage strains representing 8 new species isolated in wastewater and freshwater ponds in the Sevilla area in Spain and infecting *S. enterica* serovar Typhimurium strain ATCC 14028S. Based on genome-wide analyses independent from prior knowledge on functional annotation, we could ascribe taxonomic classification down to the genus level. These phages all belong to the *Caudoviricetes* class (dsDNA-tailed bacteriophages with an HK97 major capsid fold) and to four different genera (Figure 3 and Figure 4). *Kuttervirus* phages are overrepresented, with six strains identified, five of them being new species. Only Salmonella phage Salfasec13b, belonging to the *Lederbergvirus* genus, was predicted as a temperate phage.

We wish to highlight in our study two important methodological points. The first one is that by combining the results of four different gene callers, including Phanotate especially designed for phage genomes (but that nevertheless tends to over-predict ORFs), we believe we could predict as many ORFs as is reasonably possible. However, we also showed that manual inspection and curation are still needed to get rid of obvious false positive predictions. The second point concerns the improvement of functional annotation with PHROG, a database dedicated to viral proteins at large (bacterial and archaeal viruses and their prophages). We believe this is a promising approach for functional annotation as it relies on viral protein orthologs clustering based on remote homology detection. PHROG capitalizes on decades worth of past work of many research teams that have led to the experimental validations of many functions, and PHROG also benefits from the manual annotation of various experts in the field (we could propose in this study a functional annotation to 24 phrog clusters previously annotated with unknown function).

We mined our 10 annotated genomes for relevant functions in the context of future biocontrol application of *Salmonella* spp. We could, thus, identify phage-encoded proteins targeting bacterial anti-phage defenses such as Abortive infection, Restriction-Modification, and CRISPR-Cas systems. Our study also illustrates that in otherwise very similar genomes such as our six *Kuttervirus* (ANI values between 0.94 and 0.98), we could identify subtle variations in gene equipment that can single out one phage from its closest relatives. Indeed, we predicted that the Salmonella phage SeF3a genome alone codes for a new anti-Abi system based on a phage-encoded Phage Shock Protein A ortholog, potentially conferring Salmonella phage SeF3a a selective advantage over the other five viruses of the same *Kuttervirus* genus. This example emphasizes the need to extensively mine bacteriophage genomes in order to tailor therapeutic cocktails matching the targeted pathogen. Conversely, one has to obtain as much genomic information as possible about the pathogen itself in order to identify its anti-phage defenses and select appropriate phages to formulate a therapeutic cocktail that can overcome these cellular defenses. The recent publication by Tesson et al. is helpful in that respect with the description of DefenseFinder, a bioinformatic tool specially designed to systematically predict anti-phage systems in prokaryotic genomes [77].

The coevolution of bacteria and their viruses for ages beyond count is often described as an arms race that gifted both protagonists with highly diverse molecular weapons and shields. To go beyond empiric approaches and rationally design efficient phage-based biocontrol of bacteria, we need to adopt a holistic approach to the bacteriophage/host binomen. In that respect, genomic and functional analyses of both partners are crucial to meeting the existing challenges in tackling bacterial infections.

## Figures and Tables

**Figure 1 ijms-23-08917-f001:**
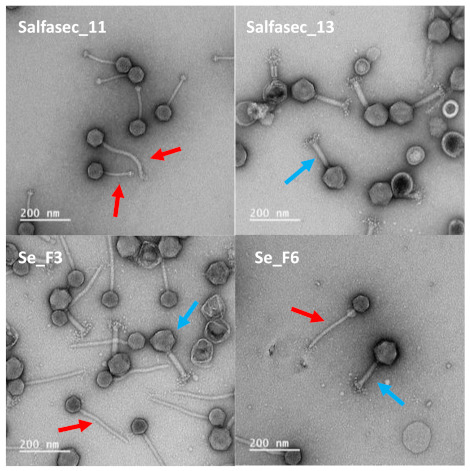
TEM images of negatively stained purified phage suspension (Salfasec_11, Salfasec_13, Se_F3, and Se_F6). Phages were purified by centrifugation of ATCC 14028S culture lysates and resuspended in TEM buffer. Grids were prepared and visualized as described in the Materials and Methods section. Red arrows: siphovirus morphology. Blue arrows: myovirus morphology. Scale bar: 200 nm.

**Figure 2 ijms-23-08917-f002:**
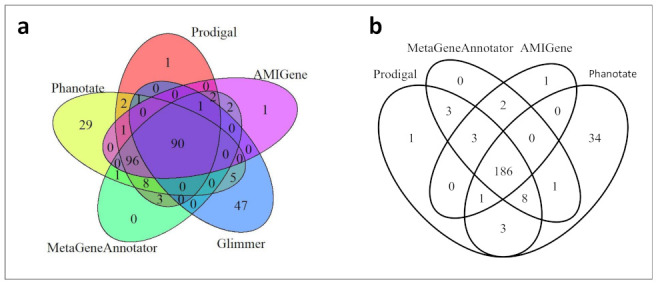
Comparison of the ORFs predictions for the Se_AO1 genome. (**a**) Venn diagram for AMIGene, Glimmer, MetaGeneAnnotator, Phanotate, and Prodigal predictions (*n* = 290 ORFs). (**b**) Venn diagram omitting Glimmer predictions (*n* = 243 ORFs).

**Figure 3 ijms-23-08917-f003:**
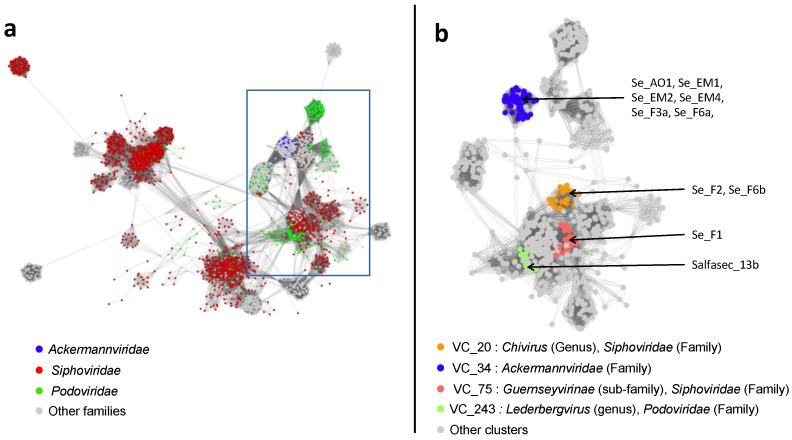
Interaction network between *Caudoviricetes* bacteriophages based on shared protein ortholog families. The interaction network was built using vContact2 and visualized with Cytoscape. Each dot corresponds to a single phage. (**a**) Network comprising all the *Caudovirictes* viruses in the vContact2 database. Viruses belonging to the *Ackermannviridae*, *Siphoviridae*, and *Podoviridae* families are highlighted in blue, red, and green, respectively. The rectangle indicates the region in the network where the 10 phages studied here are positioned. (**b**) Sub-network of the region underlined in (**a**) restricted to *Ackermannviridae*, *Podoviridae*, and *Siphoviridae* surrounding our 10 new phages. Viruses belonging to the *Chivirus* genus (VC_20), the *Ackermannviridae* family (VC_34), the *Guernseyvirinae* sub-family (VC_75), and the *Lederbergvirus* genus (VC_243) are highlighted in orange, blue, red, and green, respectively. The positions in the sub-network of the 10 new phages are indicated by arrows. *Siphoviridae* and *Podoviridae* families have been removed from the latest ICTV taxonomic classification but still existed in the vContact2 database we used for our analyses.

**Figure 4 ijms-23-08917-f004:**
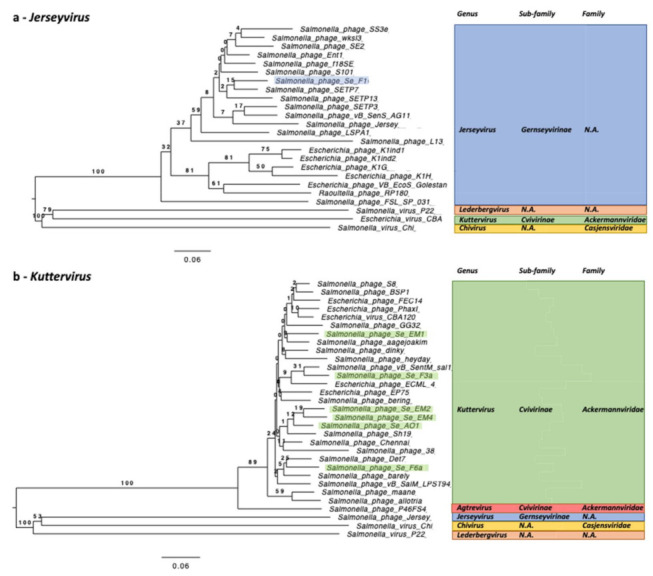
Taxonomic affiliation at the genus level based on genome-wide amino acid analysis by VICTOR. (**a**) *Jerseyvirus* genus. Twenty *Jerseyvirus* from the RefSeq database were included, as well as one *Kuttervirus* (Escherichia virus CBA120), one *Chivirus* (Salmonella phage Chi), and one *Lederbergvirus* (Salmonella phage P22) serving as outgroups. Se_F1 is highlighted in light blue. (**b**) *Kuttervirus* genus. Twenty-one *Kuttervirus* from the RefSeq database were included, as well as one *Agtrevirus* (Salmonella phage P46FS4), one *Lederbergvirus* (Salmonella phage P22), one *Chivirus* (Salmonella phage Chi), and one *Jerseyvirus* (Salmonella phage Jersey) serving as outgroups. Se_AO1, Se_EM1, Se_EM2, Se_EM4, Se_F3a, Se_F6a are highlighted in light green.

**Figure 5 ijms-23-08917-f005:**
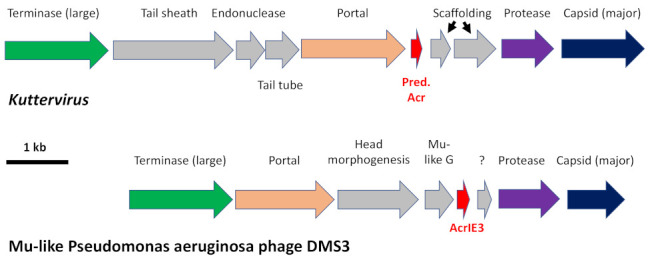
Comparison of the genomic surroundings of bona fide Acr-encoding genes (*acrIF2*) in Mu-like Pseudomonas aeruginosa phage DMS3 (accession NC_008717) and the predicted Acr-encoding ORF in our 6 *Kuttervirus*. For clarity, genes are drawn to scale but not the intergenic regions.

**Table 1 ijms-23-08917-t001:** The 10 contigs selected for our downstream genomic analyses with their closest homolog in the NCBI nr database.

Contig	Size (bp)	Genus Cluster	Blastn Best Hit ^1^	Cov. ^2^	Evalue	Identity	Length (bp) ^3^
Se_AO1	157,543	3	Salmonella phage PhiSH19	97%	0.0	99.6%	157,785
Se_EM1	159,312	3	Salmonella phage S8	93%	0.0	98.2%	158,432
Se_EM2	156,948	3	Salmonella phage S8	89%	0.0	96.1%	158,432
Se_EM4	157,810	3	Salmonella phage vB_SalM-LPST94	89%	0.0	97.6%	156,548
Se_F1	42,784	2	Salmonella phage S101	90%	0.0	92.2%	42,621
Se_F2	59,084	1	Salmonella phage BPS1	99%	0.0	98.9%	58,852
Se_F3a	157,219	3	Salmonella phage vB_SentM_sal1	97%	0.0	99.2%	157,338
Se_F6a	157,240	3	Salmonella phage Chennai	91%	0.0	96.0%	157,462
Se_F6b	59,519	1	Salmonella phage Chi	98%	0.0	93.0%	59,578
Salfasec_13b	40,455	4	Salmonella phage S149	93%	0.0	99.8%	41,391

^1^ We kept for Blastn best-hits only fully sequenced phage genomes (accession numbers in Appendix B, Table A3). ^2^ Query coverage. ^3^ Query length.

**Table 2 ijms-23-08917-t002:** Summary of PhageTerm results.

Contig	Ends	Left	Right	Permutated	Class	Type	Sequence Cohesive
Se_F1	Redundant	Random	Random	Yes	-	-	-
Se_F2	Non-redundant	48,283	48,294	No	Cos (5′)	Lambda	GGTGCGCAGAGC
Se_F3a	Redundant	Random	Random	Yes	-	-	-
Se_F6a	Redundant	Random	Random	Yes	-	-	-
Se_F6b *	Non-redundant	1	12	No	Cos (5′)	Lambda	GGTGCGCAGAGC

* Result to be considered carefully as the mean genome coverage is 15, under the software recommended coverage threshold (50). Ends: describe whether the ends are unique or redundant. Left and Right: nucleotide position of the Left and the Right termini in the DNA sequence. Permutated: describe whether individual encapsidated DNA molecules are all the same (non-permutated, e.g., for *cos* packaging) or circularly permutated (e.g., head-full packaging). Class: predicted packaging mechanism. Type: model phage for the considered packaging mechanism.

**Table 3 ijms-23-08917-t003:** Taxonomic classification of the 10 newly isolated *Salmonella* phages included in this study.

Strain	Family	Sub-Family	Genus	Species
Salmonella phage SeAO1	*Ackermannviridae*	*Civivirinae*	*Kuttervirus*	*Kuttervirus SH19*
Salmonella phage SeEM1	* Ackermannviridae *	* Civivirinae *	* Kuttervirus *	* “Kuttervirus SeEM1” *
Salmonella phage SeEM2	* Ackermannviridae *	* Civivirinae *	* Kuttervirus *	* “Kuttervirus SeEM2” *
Salmonella phage SeEM4	* Ackermannviridae *	* Civivirinae *	* Kuttervirus *	* “Kuttervirus SeEM4” *
Salmonella phage SeF3a	* Ackermannviridae *	* Civivirinae *	* Kuttervirus *	* “Kuttervirus SeF3a” *
Salmonella phage SeF6a	* Ackermannviridae *	* Civivirinae *	* Kuttervirus *	* “Kuttervirus SeF6a” *
Salmonella phage SeF1	* N.A. ^#^ *	* Guernseyvirinae *	* Jerseyvirus *	* “Jerseyvirus SeF1” *
Salmonella phage SeF2	* Casjensviridae *	* N.A. *	* Chivirus *	* “Chivirus SeF”2 *
Salmonella phage SeF6b	*Casjensviridae*	*N.A.*	*Chivirus*	*Chivirus FSLSP088*
Salmonella phage Salfasec13b	* N.A. *	* N.A. *	* Lederbergvirus *	* “Lederbergvirus Salfasec13b” *

^#^ Non-Applicable. In blue with quotation marks: new phage species.

**Table 4 ijms-23-08917-t004:** Suggested improvement to the functional annotation of 24 phrog families currently annotated “unknown function” in the PHROG database. * Functional categories defined in PHROG.

Phrog	Sequences ^#^	Proposed Phrog Annotation	Functional Category *
phrog_4858	27	head-tail_connector_protein	connector
phrog_12508	8	endopeptidase_domain-containing_protein	lysis
phrog_600	231	lysosyme	
phrog_3838	37	receptor_binding_protein	tail
phrog_1030	144	tail_needle_knob	
phrog_3833	37	base_plate_wedge_protein	
phrog_2448	60	carbohydrate-binding_protein	
phrog_3018	47	head_decoration_protein	head and packaging
phrog_33262	2	phosphoadenosine_phosphosulfate_reductase	moron, auxiliary metabolic gene and host takeover
phrog_1203	125	phage_shock_protein_A	
phrog_3614	40	complement_C1Q-like_protein	
phrog_4435	31	ryanodine_receptor	
phrog_4049	34	ABC-type_bacteriocin_transporter_peptidase_domain	
phrog_11528	9	threonine_deaminase	
phrog_2426	61	magnesium_chelatase	
phrog_4600	29	DNA-binding_protein	transcription regulation
phrog_16119	6	transcription_initiation_factor	
phrog_7401	16	DNA-binding protein	
phrog_3786	38	transcription_initiation_factor	
phrog_1934	79	DNA-binding_protein	
phrog_12574	8	Anti-TRAP_protein	
phrog_6201	20	CI-like_repressor	
phrog_37032	2	pentapeptide_repeat_protein	other
phrog_4700	29	ADP-ribose_transferase	DNA, RNA, and nucleotide metabolism

^#^ Number of orthologous protein sequences in the phrog cluster. * Functional categories defined in PHROG.

**Table 5 ijms-23-08917-t005:** Summary of the final syntaxic and functional annotations of the 10 phage genomes. “Unknown function” represents the percentage of predicted ORFs annotated “unknown function”. “Singleton” represents the percentage of predicted ORFs that do not belong to any phrog family.

Strain	Size (bp)	ORFs	tRNAs	Unknown Function ^#^	Singleton ^&^	Accession
Salmonella phage SeAO1	157,543	233	Asn, Gln, Met, Ser, Tyr, Val	57.5%	10.3%	SAMEA6862452
Salmonella phage SeEM1	159,312	236	Asn, Met, Pro, Ser, Tyr	57.2%	11.0%	SAMEA6862907
Salmonella phage SeEM2	156,948	233	Asn, Met, Pro, Ser, Tyr	55.4%	11.1%	SAMEA6862908
Salmonella phage SeEM4	157,810	231	Asn, Met, Pro, Ser, Val	55.0%	10.8%	SAMEA6862910
Salmonella phage SeF1	42,784	74	None	37.8%	8.1%	ON809761
Salmonella phage SeF2	59,084	86	None	51.2%	12.8%	ON809762
Salmonella phage SeF3a	157,219	236	Asn, Ile, Met, Tyr	55.5%	10.2%	ON809763
Salmonella phage SeF6a	15,7240	233	Asn, Ile, Pro, Ser, Tyr	53.6%	9.0%	ON809764
Salmonella phage SeF6b	59,519	84	None	50.0%	11.9%	ON809755
Salmonella phage Salfasec13b	40,455	74	Asn	41.9%	12.2%	ON809756

^#^ Percentage of predicted ORFs annotated “unknown function”. ^&^ Percentage of predicted ORFs that do not belong to any phrog family.

**Table 6 ijms-23-08917-t006:** Predicted Acr proteins in the 10 phage genomes studied.

Salmonella Phage	Genus	Phrog	Protein	Size (aa)	PaCRISPR Score	AcRanker Score	AcrHub ID ^#^	CRISPR Type ^&^
SeAO1	*Kuttervirus*	4097	Gp_007	56	0.62	−0.75	Acr00210	II-A
		519	Gp_015	71	0.54	−2.08	Acr00037	I-D
		5934	Gp_049	105	0.69	−4.64	Acr00193	II-A
		3594	Gp_088	62	0.58	−3.57	Acr00054	I-D
SeEM1	*Kuttervirus*	4097	Gp_008	56	0.62	−0.75	Acr00210	II-A
		3594	Gp_080	62	0.55	−2.90	Acr00212	II-A
SeEM2	*Kuttervirus*	4097	Gp_008	56	0.62	−0.75	Acr00210	II-A
		519	GP_016	71	0.54	−2.08	Acr00037	I-D
SeEM4	*Kuttervirus*	4097	Gp_007	56	0.62	−0.75	Acr00210	II-A
		519	Gp_015	71	0.54	−2.08	Acr00037	I-D
SeF3a	*Kuttervirus*	4097	Gp_008	56	0.62	−0.74	Acr00210	II-A
		519	Gp_014	78	0.48 *	−1.17	Acr00037	I-D
		3594	Gp_082	62	0.54	−2.77	Acr00212	II-A
SeF6a	*Kuttervirus*	4097	Gp_007	56	0.62	−0.74	Acr00210	II-A
		5934	Gp_042	110	0.68	−3.85	Acr00211	II-A
SeF1	*Jerseyvirus*	13,851	Gp_038	73	0.58	−4.11	Acr00209	II-A
		18,511	Gp_057	61	0.53	−1.96	Acr00311	V-A
		6094	Gp_059	173	0.83	−3.57	Acr00061	I-E
		2180	Gp_068	61	0.53	−4.87	Acr00128	I-F
		1402	Gp_072	69	0.64	−3.92	Acr00070	I-F
		Singleton	Gp_073	62	0.70	−4.91	Acr00264	II-C
Se_F2	*Chivirus*	7965	Gp_051	120	0.84	−4.46	Acr00254	II-C
		7734	Gp_052	111	0.54	−4.86	Acr00029	I-D
		4445	Gp_057	85	0.77	−5.17	Acr00159	II-A
Se_F6b	*Chivirus*	7965	Gp_051	120	0.84	−4.21	Acr00275	II-C
		7734	Gp_052	111	0.53	−4.80	Acr00029	I-D
		4445	Gp_058	86	0.69	−4.94	Acr00198	II-A
Salfasec_13b	*Lederbergvirus*	11,224	Gp_044	41	0.65	−4.56	Acr00064	I-E

^#^ Closest homolog in the AcrHub database. ^&^ Inhibited CRISPR type. * PaCRISPR score below 0.5, but this predicted Acr belongs to the phrog_519 that is found in the other five *Kuttervirus* genomes with reliable scores.

## Data Availability

Raw sequencing reads and Genbank files for the 10 newly isolated salmonella phages studied here are openly available at NCBI under the BioProject accession numbers PRJNA767534 and PRJEB37792. Input and output files pertaining to vContact2 analyses are publicly available since 17 June 2022 at figshare (https://doi.org/10.6084/m9.figshare.20088356). Output files pertaining to VICTOR analyses are publicly available since 17 June 2022 at figshare (https://doi.org/10.6084/m9.figshare.20088416 for *Kuttervirus* and https://doi.org/10.6084/m9.figshare.20088473 for *Jersyvirus*).

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
