# Peer review of "Genetic Mining of Newly Isolated Salmophages for Phage Therapy"

_ijms, 2022, doi:10.3390/ijms23168917_

Round 1
Reviewer 1 Report
My sympathies with the language and style, clear outline of the results, and overall presentation of the subject.
This is thorough and detailed analysis of a set of Salmonella phages on the genome level using a set of bioinformatics tools allowing to perform "Syntaxic" annotation of phage genomes and deciphering a good amount of unknown functions of genes. The taxonomic affiliation has been described on genus level for 10 investigated Salmonella phages. In the light of prospective phage therapy special attention is devoted to the identification of the presence of anti-defense systems in investigated phages, e.g. presence of anti-CRISPR proteins. Minor grammar issues: 21. - characterized 127.- Lederberg 308. - "taxonomy" -unnecessary 309. - "of" needed 630.- anti-host I found this manuscript a valuable source of new information with a good level of discussion on several aspects of phage genome analysis.Author Response
Reviewer 1 (please see also attachment)
My sympathies with the language and style, clear outline of the results, and overall presentation of the subject.
This is thorough and detailed analysis of a set of Salmonella phages on the genome level using a set of bioinformatics tools allowing to perform "Syntaxic" annotation of phage genomes and deciphering a good amount of unknown functions of genes. The taxonomic affiliation has been described on genus level for 10 investigated Salmonella phages. In the light of prospective phage therapy special attention is devoted to the identification of the presence of anti-defense systems in investigated phages, e.g. presence of anti-CRISPR proteins. Minor grammar issues: 21. - characterized 127.- Lederberg 308. - "taxonomy" -unnecessary 309. - "of" needed 630.- anti-host I found this manuscript a valuable source of new information with a good level of discussion on several aspects of phage genome analysis.
All 5 grammar issues have been corrected in the revised manuscript.

Reviewer 2 Report
The paper is very well written in terms of grammar and use of language. It is also well laid out in terms of the logic of the story. However, it is overly long and some instances editing for brevity would be useful to allow the reader to identify the main points. It is 25 pages for what amounts to a genome description of 10 phages. While it is very well carried out and very thorough, it is overly long.
Eg Line 145 -153 . This can be vastly reduced to convey the important information – which is the number of phage isolates sequenced. The details of who isolated and trimming of reads etc are superfluous. A simple citation of where the phage came from will suffice
The major issues are the availability of genome sequences and the taxonomy. Both of which are easily solved. Until the genome sequences are made public this work should not be published. Given the extensive efforts of the authors to provide all the other data via figshare etc, I suspect this is a lag in the database. But please provide accession numbers for each phage. the read data is present
The taxonomy of these phages needs to be updated for the most recent terms and ensure they classified at the correct level
Minor
L171 – L185
The authors seem to be suggesting that the similar genomes are from cross-contamination rather than isolation of the same phage multiple times (which can occur), which is incredibly honest of them. It is not clear what the case is ?
It is also not clear if these mixed isolates have now been purified to single phage – can the contaminant be purified out ?
Line 185 – why 99% is chosen for a species is not clear. There are clearly defined limits of >95% ANI for phage to be called a species (the authors cite the work later). Changing the cut-off for this work is not helpful for consistency, without any justification of why such a high value should be used.
This is easily solved by the use of the word “strain” . However what is not, is why the other genomes were not looked at any further and disregarded for further analysis. Using the deREP and mash based analysis will give the rapid ANI values. However, this approach will miss single SNP, which will differentiate between phages. If a phage is 99% ANI then, it clearly is a different phage- it might be the same “strain”.
If these were from individuals isolates, which they appear to be, then they may contain useful information of the micro-diversity of phage.
As the metadata (time/location/sources/etc) of these phages is not included in the paper, it is not possible to determine if geographically isolated phages are identical or 99% similar. There are many other reports of identical or very similar phage isolates being sequenced and isolated independently. If the authors do not wish to include them in their analysis, that is their prerogative. But if they have 1 SNP different, they are different phages. Based on the data presented this is not possible to determine. Please clarify this
But adding the genomes and reads to a public database would be valuable resource to the community interested in phage genomic diversity.
238 citation to support this common knowledge
L300-302 . citations for this statement of fast evolving genomes are required for phages. Given the limited literature on rate of evolution in phages , some of these are very bold statements. The calculated rate of evolution on cyanophages does not support this statement for example (PMC6722890) or work on Lactococcus phages.
L297-328 .I would suggesting editing for brevity as much of this information is not required to understand the results
There are some issues that need to be addressed within the Taxonomy section.
The Siphoviridae/Podoviridae no longer exist as families and should not be used. The latest families need to used and these families removed
Table 4 incorrectly list the phage as species. These are not species they are the name of phage isolates. Phage species are named based on freeform binomials. So if they do turn out to be new species, they need to be named in this format eg “Kuttervirus Se_AO1” that conforms with ICTV guidelines. Please consults the ICTV website for full details. On use of italics and “” for taxonomy that is provisional
In addition, it is not clear if these are all new species or similar to existing species or would cluster together. It should be determined if these are new species based on the well defined ICTV guidelines of 95% identity. Please state the existing species if they are or identify that they are a likely new species. VIRIDC is an excellent tool for rapidly being able to do this although other methods work as well. Please also state if the 70% identity across the genome is met for inclusion within an existing genus – this is not clear from the data presented.
Instead, or as well as guest star. Please state the closest phage species in binomial form
L377 – the taxonomic classification supports rather than confirming the PhageTerm results
L386-435 Again editing this section for brevity would improve the manuscript
L542-543 . It is not clear what is meant by “family” in terms of phage, is this taxonomic family ?
. More importantly the justification of not using similar phages is not entirely clear. Phages are combined in a cocktail for multiple reasons, and these are not reliant on co-infection, which will be altered by the imm protein. These reasons include the synergistic effects eg a phage with a depolymerase that exposes the receptor to help the other phage infect. Or simply because a single phage does not target the diversity of bacteria present and thus multiple phages are required. In these cases, having phages of the same “family” has benefit. It will the RBP that is of importance and not the imm protein. This statement should be toned down and the nuance of the situation explained
L668 – there work has a somewhat abrupt ending. A summary or conclusion of the work would help . Apologies, this is later. It was the format of the journal I was not familiar with.
L670 – please provide metadata for the phage isolates. Place of isolation /date . Accession number for each phage
L809-810 the LOCUS tags do not conform to the internationals standards. Which is all uppercase letters see https://ena-docs.readthedocs.io/en/latest/faq/locus_tags.html is this a contributing factor as to why the genomes have not been released ?
L815 – it is not clear if the genomes have been reordered or just the position at which ORFS are numbered ? Given the authors have gone to considerable length to use PhageTerm to identify the termin of the genome . When these are clearly defined termini , the genome should be orientated to reflect this .Else why identify the termini if the genomes are not going to be orientated .
This may have been done , but it is not clear and with no accession numbers there is no way to check
L817 – Would suggest for brevity the section is edited to provide only the details required. Which is the software settings and version.
L922 – Caudovirales has been removed. Please consult ICTV which provides the latest taxonomy . These are now within the Caudoviricetes https://talk.ictvonline.org/taxonomy/
Author Response
Reviewer 2 (please see also attachment)
The paper is very well written in terms of grammar and use of language. It is also well laid out in terms of the logic of the story. However, it is overly long and some instances editing for brevity would be useful to allow the reader to identify the main points. It is 25 pages for what amounts to a genome description of 10 phages. While it is very well carried out and very thorough, it is overly long.
Eg Line 145 -153. This can be vastly reduced to convey the important information – which is the number of phage isolates sequenced. The details of who isolated and trimming of reads etc are superfluous. A simple citation of where the phage came from will suffice
Paragraph reduced accordingly.
The major issues are the availability of genome sequences and the taxonomy. Both of which are easily solved. Until the genome sequences are made public this work should not be published. Given the extensive efforts of the authors to provide all the other data via figshare etc, I suspect this is a lag in the database. But please provide accession numbers for each phage. the read data is present
The taxonomy issues have been corrected as mentioned below. However, the attribution of the final accession numbers is still pending. We submitted our genomes to GenBank (submission ID: 2595291) on June 20th 2022. Accession numbers were attributed by GenBank on June 21st 2022 (see email attached). We have inquired at GenBank about the status of our submitted genome as they do not appear in the related NCBI account under the BioProject number PRJNA767534. If there is an issue with our submission, we haven't been informed by NCBI. We modified Table 5 (formerly Table 6) in the revised manuscript by adding these numbers, hoping they will soon lead to a proper GenBank file.
The taxonomy of these phages needs to be updated for the most recent terms and ensure they classified at the correct level
During the writing of the manuscript submitted at IJMS, the authors extensively used the ICTV website (the "old" version) to ensure correctness of the taxonomic information. However major changes occurred recently in the Caudoviricetes class (2021.015B proposal, ratified on March 2022) and the new taxonomy was seemingly made available online on the ICTV website shortly before or after we wrote our manuscript. We are of course willing to correct the taxonomic information contained in this manuscript (mainly the disappearance of the Siphoviridae and Podoviridae families as well as the creation of the Casjensviridae family). Figure 4 (formerly Figure 5) and Table 3 (formerly Table 4) were also corrected accordingly. We also corrected the typography when required by the ICTV naming rules.
However, one noticeable exception will be made in Figure 3 as the vContact2 analysis relies on a database built by Bolduc et al. before the release of the new taxonomy. Hence Podoviridae and Siphoviridae families are still mentioned in Figure 3. However, we inserted in the text the following warning in the Figure 3 caption:
"Siphoviridae and Podoviridae families have been removed from the latest ICTV taxonomic classification but were still existing in the vContact2 database we used for our analyses."
Minor
L171 – L185
The authors seem to be suggesting that the similar genomes are from cross-contamination rather than isolation of the same phage multiple times (which can occur), which is incredibly honest of them. It is not clear what the case is?
To be completely honest, we do not know for sure … I am rather inclined myself to think we isolated the same virus several times as we had in other unrelated projects in the team with different environmental samples. Despite being a phage lab, we extremely rarely observed cross-contaminations at the bench.
It is also not clear if these mixed isolates have now been purified to single phage – can the contaminant be purified out?
These mixed isolates have not yet been purified to single phage but will be as soon as we decide which single phages are interesting for our downstream projects. To do so, after several rounds of re-isolation, we will screen by PCR several isolates to identify the phage and verify the purity of the preparation.
Line 185 – why 99% is chosen for a species is not clear. There are clearly defined limits of >95% ANI for phage to be called a species (the authors cite the work later). Changing the cut-off for this work is not helpful for consistency, without any justification of why such a high value should be used. This is easily solved by the use of the word “strain”. However what is not, is why the other genomes were not looked at any further and disregarded for further analysis. Using the deREP and mash based analysis will give the rapid ANI values. However, this approach will miss single SNP, which will differentiate between phages. If a phage is 99% ANI then, it clearly is a different phage- it might be the same “strain”. If these were from individuals isolates, which they appear to be, then they may contain useful information of the micro-diversity of phage.
We acknowledge that the use of improper terms for taxonomic description or missing information required some revisions in the manuscript. Following Reviewer 2 advices, we used VIRIDIC instead of dRep 1) to determine which of the 18 contigs assembled represent potential novel species or not and 2) to determine after genus assignment which phage strains are truly novel species within the considered genus according to the current ICTV classification. We sticked to the recommended threshold for species and genus demarcations. Table 1 was suppressed and Table 2 (Table 1 in the revised version) modified to integrate VIRIDIC results. This simplifies (a bit) the message and the reading for the lector. "Species" was replaced by "Strain" when necessary and Table 3 (formerly Table 4) rebuilt to aggregate VIRIDIC results allowing to identify novel from existing species. A (brief) paragraph was in the Material and Methods section for VIRIDIC usage.
As the metadata (time/location/sources/etc) of these phages is not included in the paper, it is not possible to determine if geographically isolated phages are identical or 99% similar. There are many other reports of identical or very similar phage isolates being sequenced and isolated independently. If the authors do not wish to include them in their analysis, that is their prerogative. But if they have 1 SNP different, they are different phages. Based on the data presented this is not possible to determine. Please clarify this. But adding the genomes and reads to a public database would be valuable resource to the community interested in phage genomic diversity.
We chose in this study to focus on potentially new phage species, although we are well aware that tiny differences between two strains are interesting in the light of microdiversity of phages. The authors will deposit the genomes of the strains not considered for functional annotation in GenBank.
238 citation to support this common knowledge
The authors acknowledge the lack of proper reference(s) about a much-debated subject. We thus added in the manuscript a reference to a very recently published paper with a humoristic title (for "The Lord of the Rings" fans) dealing with this matter.
Dimonaco et al. (2022) No one tool to rule them all: prokaryotic gene prediction tool annotations are highly dependent on the organism of study. Bioinformatics, 38(5), 1198-1207.
L300-302 . citations for this statement of fast evolving genomes are required for phages. Given the limited literature on rate of evolution in phages, some of these are very bold statements. The calculated rate of evolution on cyanophages does not support this statement for example (PMC6722890) or work on Lactococcus phages.
We agree that this sentence should be revised. We had in mind phage training such as described by Borin et al. (PMID 34083444) where co-evolution of phage Lambda with its host for 28 days "rapidly" led to resistant phage strains. "Rapidly" must be tempered as 28 days represent about 1,344 bacterial generations at 37°C and in vitro. It is likely that conditions favoring the rapid emergence of advantageous phenotypic traits for the phage are not often met in physiological conditions. This study however highlights the various pathways leading to beneficial genetic mutations, such as acquisition of genetic material by recombination with host prophages. In the light of the literature cited by Reviewer 2, the sentence has been suppressed the revised manuscript, the rate of evolution of phages being a subject in itself that cannot be treated in one sentence and is beyond the scope of the current paper!
L297-328 .I would suggesting editing for brevity as much of this information is not required to understand the results.
This paragraph has been shortened as advised.
There are some issues that need to be addressed within the Taxonomy section. The Siphoviridae/Podoviridae no longer exist as families and should not be used. The latest families need to used and these families removed.
These issues have been corrected with the exception of Figure 3 for the reasons mentioned above.
Table 4 incorrectly list the phage as species. These are not species they are the name of phage isolates. Phage species are named based on freeform binomials. So if they do turn out to be new species, they need to be named in this format eg “Kuttervirus Se_AO1” that conforms with ICTV guidelines. Please consults the ICTV website for full details. On use of italics and “” for taxonomy that is provisional.
Table 4 (Table 3 in the revised manuscript) has been modified accordingly.
In addition, it is not clear if these are all new species or similar to existing species or would cluster together. It should be determined if these are new species based on the well defined ICTV guidelines of 95% identity. Please state the existing species if they are or identify that they are a likely new species. VIRIDC is an excellent tool for rapidly being able to do this although other methods work as well. Please also state if the 70% identity across the genome is met for inclusion within an existing genus – this is not clear from the data presented. Instead, or as well as guest star. Please state the closest phage species in binomial form.
As mentioned above, we provide in the revised manuscript VIRIDIC analyses allowing to identify new species or existing ones in the current ICTV taxonomy.
L377 – the taxonomic classification supports rather than confirming the PhageTerm results
The sentence was corrected in the revised manuscript and we seize the opportunity to shorten the paragraph.
L386-435 Again editing this section for brevity would improve the manuscript
This paragraph has been shortened as advised.
L542-543 . It is not clear what is meant by “family” in terms of phage, is this taxonomic family? More importantly the justification of not using similar phages is not entirely clear. Phages are combined in a cocktail for multiple reasons, and these are not reliant on co-infection, which will be altered by the imm protein. These reasons include the synergistic effects eg a phage with a depolymerase that exposes the receptor to help the other phage infect. Or simply because a single phage does not target the diversity of bacteria present and thus multiple phages are required. In these cases, having phages of the same “family” has benefit. It will the RBP that is of importance and not the imm protein. This statement should be toned down and the nuance of the situation explained.
"Family" was incorrectly used in this paragraph. After reading Reviewer 2 comment, we decided to remain factual and simplify the text.
L668 – there work has a somewhat abrupt ending. A summary or conclusion of the work would help . Apologies, this is later. It was the format of the journal I was not familiar with.
Acknowledged.
L670 – please provide metadata for the phage isolates. Place of isolation /date . Accession number for each phage
Accession numbers assigned by GenBank have been added to Table 6 (Table 5 in the revised manuscript). As we do not have homogenous metadata for all phage isolates, we chose to give only general information (country, city, type of water) in the Materials and Methods section.
L809-810 the LOCUS tags do not conform to the internationals standards. Which is all uppercase letters see https://ena-docs.readthedocs.io/en/latest/faq/locus_tags.html is this a contributing factor as to why the genomes have not been released ?
It may have been so. It will be either corrected by GenBank or we will submit a second version of our annotated genomes to comply to these rules.
L815 – it is not clear if the genomes have been reordered or just the position at which ORFS are numbered? Given the authors have gone to considerable length to use PhageTerm to identify the termin of the genome. When these are clearly defined termini, the genome should be orientated to reflect this. Else why identify the termini if the genomes are not going to be orientated .This may have been done , but it is not clear and with no accession numbers there is no way to check
For the sake of genomic comparison between the modules comprising the structural genes (a study not related to the present work), we chose to orientate all our genomes starting with the terminase small subunit-encoding gene (gene product 1). The position of the termini for Se_F2 and Se_F6b are indicated in Table 4. For Se_F6b the predicted terminus start at the first nucleotide of the sequence and we initially chose to leave Se_F2 in its current orientation to ease our comparisons between Se_F2 and Se_F6b. As a second version of all our genomes is probably required, we will reorientate Se_F6b genome starting with the predicted terminus in the updated version.
L817 – Would suggest for brevity the section is edited to provide only the details required. Which is the software settings and version.
Text shortened as advised.
L922 – Caudovirales has been removed. Please consult ICTV which provides the latest taxonomy . These are now within the Caudoviricetes https://talk.ictvonline.org/taxonomy/
This issue has been dealt with as mentioned above.

Reviewer 3 Report
The authors described the sequencing and annotation of 10 newly isolated phages infecting Salmonella enterica, and then explore their usefulness to phage therapy by looking at the anti-defense system these phages encode. They use several bioinformatic tools to detect orf and assign them functions. In a general manner the text is well written and clear, and the content is highly sufficient for publication. The interest in phage therapy brings an interesting aspect to this annotation paper that I quite appreciated. I only have a few minor comments:
- Figure 1 could use some arrow to point as the phage described in the text.
- Table 1 is a bit hard to read in the present form. The color coding is not explained. I think a gradient of color going with the pairwise values would help to read the figure without having to read each value and guide the lector to understand the groups the authors speak about in the text.
- Table 3 is a bit destabilizing at first glance, and it took me some time to understand what the columns were about after reading the text, a description of the columns in the legend might help.
- In line 293, the authors state: “we kept the longest version of the ORF”. Wouldn’t it be more accurate to look if each of these versions have a RBS before the start codon, which would indicate which start is the proper one.
- Lines 515 to 517: The authors make a huge focus of anti-CRISPR proteins, but they do not speak about the prevalence of CRISPR genes in Salmonella. Is it the main defense system in this species?
- Line 527: “Of note, Salmonella phage Se_F1 genome contains four EcoB and four EcoK sites”. This would be relevant only if the author shows that these restriction enzymes from E. coli are also present in Salmonella.
- More of a curiosity side note, did the authors look at DNA modifications for these phages that also protects from multiple bacterial defenses?
- Line 870 and 897: problem with the references.
- Line 918: “new phages isolated” should be replaced by “newly isolated phages” as the phage are not new, they were just undiscovered.
Author Response
Reviewer 3 (please see also attachment)
The authors described the sequencing and annotation of 10 newly isolated phages infecting Salmonella enterica, and then explore their usefulness to phage therapy by looking at the anti-defense system these phages encode. They use several bioinformatic tools to detect orf and assign them functions. In a general manner the text is well written and clear, and the content is highly sufficient for publication. The interest in phage therapy brings an interesting aspect to this annotation paper that I quite appreciated. I only have a few minor comments:
- Figure 1 could use some arrow to point as the phage described in the text.
Figure 1 was meant as an illustration of our samples' heterogeneity. A formal identification of the phages is not possible on theses electronic micrographs. We nevertheless added colored arrows to point myovirus and siphovirus morphologies.
- Table 1 is a bit hard to read in the present form. The color coding is not explained. I think a gradient of color going with the pairwise values would help to read the figure without having to read each value and guide the lector to understand the groups the authors speak about in the text.
This Table as well as the accompanying text has been substantially modified following Reviewer 2 and 3 comments. We believe it is now much easier to read in the revised manuscript.
- Table 3 is a bit destabilizing at first glance, and it took me some time to understand what the columns were about after reading the text, a description of the columns in the legend might help.
The legend of Table 3 (Table 2 in the revised manuscript) was completed accordingly.
- In line 293, the authors state: “we kept the longest version of the ORF”. Wouldn’t it be more accurate to look if each of these versions have a RBS before the start codon, which would indicate which start is the proper one.
Our bioinformatic pipeline does not include proper RBS prediction that would allow a certain confidence in the position of the initiation codon. In some cases, the RBS is difficult to predict. We acknowledge that choosing the longest predicted ORF is not the ideal situation. For Se_AO1, 30% of the predicted ORF have a putative alternate initiation codon. We expect the rapid development of Machine Learning to be able to provide in the near future more accurate prediction tools to incorporate in phage genome bioinformatic analysis pipelines.
- Lines 515 to 517: The authors make a huge focus of anti-CRISPR proteins, but they do not speak about the prevalence of CRISPR genes in Salmonella. Is it the main defense system in this species?
Salmonella enterica species possess a type I-E CRISPR system with a cas operon and two CRISPR arrays (Kushwaha et al.). This strain also harbors a BREX-like anti-phage defense system (Zaworski et al.) for which no phage-encoded inhibitor has been identified to date. For the sake of the answer to Reviewer 3's comment, we ran DefenseFinder on Salmonella enterica 14028S to predict additional anti-phage defense systems (Tesson et al., Abby et al.). In addition to CRISPR-Cas and BREX, two PARIS (Rousset et al.), one Retron (Millman et al.), one Lamassu-Fam (Doron et al.), three restriction-modification and one Mokosh systems (Millman et al.) could be predicted. To sum-up, Salmonella enterica 14028S seems to be well equipped in various anti-phage (or more generally speaking anti-invading DNA) systems. But this is the case for many bacterial genomes as we keep discovering new anti-phage systems, few of which been characterized with even fewer phage-encoded inhibitors of these systems identified.
A sentence with the Kushwaha et al. reference was added at the beginning of paragraph 2.7.5.
- Line 527: “Of note, Salmonella phage Se_F1 genome contains four EcoB and four EcoK sites”. This would be relevant only if the author shows that these restriction enzymes from E. coli are also present in Salmonella.
Nagaraja et al. in their 1985 article (see reference below) identified in Salmonella typhimurium a type I restriction enzyme whose sequence resemble the ones found for EcoB, EcoK and EcoA type I restriction enzymes in E. coli. We haven't been specific enough in our manuscript. We propose the following sentence based on the data from Nagaraja et al.:
"Of note, Salmonella typhimurium codes for SB, an enzyme similar to E. coli EcoA, EcoB and EcoK type-I restriction enzymes [Nagaraja et al.]. Salmonella phage Se_F1 genome contains 6 restriction sites recognized by SB."
V Nagaraja, J C Shepherd, T Pripfl, T A Bickle (1985) Two type I restriction enzymes from Salmonella species. Purification and DNA recognition sequences. J Mol Biol, 182(4):579-87. PMID: 2989535 DOI: 10.1016/0022-2836(85)90243-8
- More of a curiosity side note, did the authors look at DNA modifications for these phages that also protects from multiple bacterial defenses?
In the final functional annotation of the Chivirus phages Se_F2 and Se_F6b, one can find a DNA modification-related enzyme, a DNA methyltransferase (probably a N-6-adenine-methyltranfserase) belonging to phrog_111. An ortholog is also found in Salmonella virus Chi. This finding suggests that this DNA methyltransferase is common in Chivirus phages. To the best of our knowledge, no one has ever published on the eventual chemical modifications of Chivirus phage DNA and we ourselves haven't done it. The existence of a DNA methyltransferase encoded in Chivirus genomes only suggests that viral DNA may be chemically modified by a phage-encoded enzyme. But this enzyme may also serve to modify host DNA. We did not find in the other genomes annotated in our study genes coding for other DNA-modification enzymes.
- Line 870 and 897: problem with the references.
Both problems have been solved in the revised manuscript.
- Line 918: “new phages isolated” should be replaced by “newly isolated phages” as the phage are not new, they were just undiscovered.
The sentence has been corrected as suggested. Indeed, we scientist must keep a certain humility as we definitively did not invent theses phages!
